# Efficiently Democratizing Medical LLMs for 50 Languages via a Mixture of Language Family Experts

**Guorui Zheng**[†]**, Xidong Wang**[†]**, Juhao Liang, Nuo Chen, Yuping Zheng, Benyou Wang**[*]
The Chinese University of Hong Kong, Shenzhen
`https://github.com/FreedomIntelligence/ApolloMoE`

## Abstract

Adapting medical Large Language Models to local languages can reduce barriers to accessing healthcare services, but data scarcity remains a significant challenge, particularly for low-resource languages. To address this, we first construct a high-quality medical dataset and conduct analysis to ensure its quality. In order to leverage the generalization capability of multilingual LLMs to efficiently scale to more resource-constrained languages, we explore the internal information flow of LLMs from a multilingual perspective using Mixture of Experts (MoE) modularity. Technically, we propose a novel MoE routing method that employs language-specific experts and cross-lingual routing. Inspired by circuit theory, our routing analysis revealed a *"Spread Out in the End"* information flow mechanism: while earlier layers concentrate cross-lingual information flow, the later layers exhibit language-specific divergence. This insight directly led to the development of the Post-MoE architecture, which applies sparse routing only in the later layers while maintaining dense others. Experimental results demonstrate that this approach enhances the generalization of multilingual models to other languages while preserving interpretability. Finally, to efficiently scale the model to 50 languages, we introduce the concept of *language family* experts, drawing on linguistic priors, which enables scaling the number of languages without adding additional parameters.

## 1 Introduction

The development of medical large language models (LLMs) holds great promise in addressing global healthcare inequalities (Yan et al., 2023). By democratizing access to expert knowledge, LLMs can help mitigate disparities in resource availability within healthcare systems worldwide (Tariq et al., 2020). A critical aspect of ensuring this accessibility is the inclusion of local languages, which can significantly reduce barriers to adoption and foster more equitable healthcare services (Dai et al., 2024; Permanyer et al., 2023).

However, despite the potential of multilingual LLMs in healthcare, significant challenges persist. A major obstacle is the scarcity of medical data in many languages, limiting model development for underrepresented populations. While some research has made strides in addressing this issue, these efforts often focus on only a few dominant languages, typically fewer than six (Wang et al., 2024; Qiu et al., 2024). To address this gap, we have expanded the dataset to include 12 high-resource languages, thereby improving population representation and rigorously evaluating the dataset's quality and scalability.

---

[*]Benyou is the corresponding author (*wangbenyou@cuhk.edu.cn*); [†] means contributing equally.

Expanding the coverage from 12 high-resource languages to include low-resource languages presents greater challenges due to data scarcity. Addressing this issue requires analyzing the internal information flow of large language models from a multilingual perspective to develop a generalizable approach. While some studies employ neuron analysis to investigate this (Tang et al., 2024), they typically focus on fewer than seven languages, and the complexity of neuron decomposition limits the exploration of relationships between languages and the scalability of applications. To enhance the understanding of these mechanisms, we leverage the modularity of Mixture of Experts models and introduce Hybrid-$k$ routing. This method not only ensures the activation of language-specific experts but also achieves performance on par with the vanilla top-$k$ approach, balancing interpretability and performance.

Inspired by the theory of *Circuits* (Olah et al., 2020a), we regard experts as nodes, and consider the directed acyclic graph formed by the information transmission of each token from shallow to deep layers as "Circuits". Through observation and analysis, we identify the ***"Spread Out in the End"*** mechanism in the information flow circuits, where cross-lingual integration occurs in the early layers, while language-specific differentiation happens in the later layers. This insight suggests that employing the **Post-MoE** architecture, where the MoE structure is applied only in the later layers. Experiments demonstrates that the performance of 12 high-resource languages remained stable, while low-resource languages improved even without additional training.

Building on above foundation, we further explore an efficient approach to extend the model's multilingual medical capabilities to 50 languages. Leveraging linguistic priors, we group languages into *language families*, reducing the number of expert layers required for multilingual expansion from 50 to 7. Through extensive experiments with models of 0.5B, 1.5B, and 7B parameters, we demonstrate the scalability of this method and introduce the Apollo-MoE series[1]. The series demonstrates significant potential for expansion to more languages across 50 languages, allowing for the continued increase in the number of languages without the need for additional parameters while maintaining multilingual generalization.

The main contributions are as follows: 1) We construct a high-quality medical *dataset* encompassing 12 high-resource languages, quality of which is validated by experiments. 2) We propose a new *circuits-based paradigm* for interpreting routing in a multilingual context. Through circuit analysis, we identify the *"Spread Out in the End"* mechanism. 3) By introducing **language family** experts, we efficiently extend medical LLMs to **50** languages, demonstrating its potential for scaling to more languages.

## 2  A PRELIMINARY SCALING TO 12 LANGUAGES

We commence our work with data collection. Sec. 2.1 will outline the philosophy and pipeline for collecting and processing data, while Sec. 2.2 will address quality checks and ablation studies related to data construction.

### 2.1  DATA COLLECTION AND PROCESSION

**Data Collection**  According to the key sources from which doctors and medical students acquire knowledge, we categorize these into seven valuable sources: Books, Papers, Encyclopedias, Doctor-Patient Dialogues, Exams, Websites, and Practical Guidelines, ensuring data quality from the outset. Additionally, we include general instruction tuning data to maintain foundational skills, as well as Math and Code data to enhance reasoning capabilities. Utilizing these sources, we gather high-quality data under **open source licenses** from the Internet across 12 languages, selected based on population coverage. For specific collection sources of the dataset, please refer to App. A.1.

**Data Processing**  Inspired by recent work on data construction during the Instruction Tuning phase (Cheng et al., 2024; Yue et al., 2024; Chen et al., 2023a), we employ ChatGPT[2] to transform text into question-answer

---

[1]Named after Apollo, the Greek god of medicine and light

[2]gpt-3.5-turbo-16k-0613

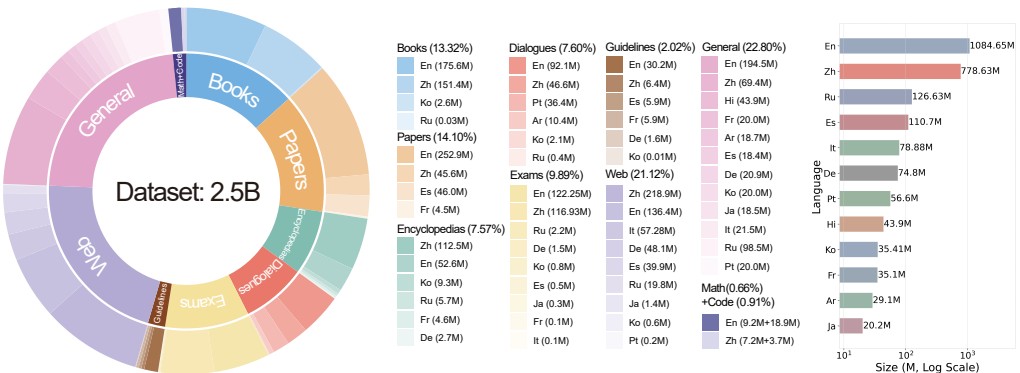

Figure 1: Taxonomy and Token statistics of Training Dataset.

pairs, enhancing the quality of our dataset. For **data leakage checks**, we adhere to the detection strategy outlined in Med-PaLM2 (Singhal et al., 2023). Specifically, if an entire question or at least 64 consecutive characters overlapped with any data item, we classify that data item as a leakage instance. In our examination of the exam data sources, we began with 621,291 exercises, from which we removed 3,479, yielding a screening rate of 0.56%. For other data sources, the filtering ratio was less than 0.01%. Ultimately, we compile a multilingual medical training set containing 2.5 billion tokens, with statistical data for various languages and sources illustrated in Fig.1. More detailed information on data processing and relevant prompts can be found in App. A.2.

**Evaluation Setup** To ensure the validity of the benchmark, we utilize the multilingual medical benchmark that is publicly available and peer-reviewed. For evaluating languages with limited resources, we follow the multilingual evaluation methodology of Llama3 (Dubey et al., 2024), employing Google Translate to translate the questions and answers of MMLU (Hendrycks et al., 2020). We also further verified the effectiveness of Google Translate in related aspects through experiments detailed in App.A.6. We employed a random selection of 3-shot queries to pose questions to the model, followed by answer extraction and evaluation of the model's responses. For detailed information regarding the composition of the evaluation set and the evaluation sample used, please refer to the App. A.3.

## 2.2 EXPERIMENTAL RESULTS

While we ensure data quality during source evaluation, we also perform a statistical analysis of the dataset's quality and composition. First, we conduct monolingual training, which involves training the model solely on data from a specific language to assess dataset's **quality** in that language. Next, we perform an ablation study focused on the **function of code and math** to determine the necessity of including this data. Multilingual training, which incorporates all available data using a random sampler, serves as our default reference. We select Gemma-2b (Team et al., 2024a) as our base model due to its moderate number of parameters. The evaluation setting and training setting used in this section are consistent with other experiments, see App. B.

**Results** As shown in Tab. 1, our **quality checking** indicates that models trained separately with language-specific data exhibit improved performance on corresponding tests, further confirming the high quality of the dataset in each language. Regarding the inclusion of **math and code data**, the model experiences an average performance loss of 5.9% when trained without this data compared to training with the full dataset, underscoring the significance of math and code for model performance. This enhancement may be attributed to the ability of math and code data to strengthen the model's reasoning capabilities. Additionally, numerals and coding language, as common elements across languages, likely serve as anchor points in multilingual

Table 1: Monolingual (■) refers to the accuracy of models trained on single language data and evaluated on the respective language evaluation set. Multilingual (■) indicates the accuracy of models trained and evaluated on all languages' datasets. The -Math&Code results reflect performance when training on all language data (■) excluding math and code data. *Avg.* denotes the average score across languages.

| Model | #params | Avg. | Ar | De | En | Es | Fr | Hi | It | Ja | Ko | Pt | Ru | Zh |
|-------|---------|------|----|----|----|----|----|----|----|----|----|----|----|----|
| Gemma | 2B | 28.1 | 18.2 | 27.6 | 37.6 | 31.8 | 23.1 | 25.7 | 26.1 | 21.2 | 25.2 | 23.1 | 49.6 | 28.0 |
| +Monolingual | 2B | - | 27.8 | 34.9 | 52.1 | 39.2 | 27.7 | 27.6 | 28.6 | 23.4 | 29.9 | 28.3 | 54.7 | 60.6 |
| +Multilingual | 2B | **48.6** | **43.7** | **50.7** | **57.8** | **48.1** | **44.5** | **40.7** | **45.2** | **43.0** | **45.1** | **40.6** | **63.7** | **60.6** |
| -Math&Code | 2B | 42.7 | 36.1 | 39.4 | 51.2 | 43.7 | 39.3 | 35.4 | 42.0 | 35.4 | 40.5 | 32.1 | 57.0 | 60.0 |

training, facilitating mutual alignment of language distributions. We also utilize the dataset to **train models of various architectures and sizes** to further validate its effectiveness, with related details and results presented in App. A.4.

# 3 SCALING WITH MoE AND ITS ROUTING ANALYSIS

This section leverages Mixture of Experts (MoE) to scale medical LLMs for better efficiency and extensibility. Sec. 3.1 introduces a new routing method and its experimental validation. Sec. 3.2 delves into a detailed analysis of the routing mechanisms using *circuits*; where we observe the *"Spread Out in the End"* phenomenon: expert routing paths are shared among languages in early layers and diverge in later layers. Inspired by the phenomenon, Sec. 3.3 presents a MoE variant called *"Post-MoE"*, which restricts routing to the later layers.

## 3.1 A LANGUAGE-SPECIFIC HYBRID ROUTING

While Sec. 2 utilizes a dense model to integrate multilingual data and extend the medical model to 12 languages, this approach presents efficiency limitations. To address these challenges, we adopt a sparse Mixture of Experts (MoE) model for a better balance between effectiveness and efficiency. The modular and functional nature of the Experts in the MoE framework is particularly advantageous in further scalability, which benefit enabling the effective expansion to more languages (especially for low-resource languages in Sec. 4.

### 3.1.1 THE PHILOSOPHY OF HYBRID-$k$

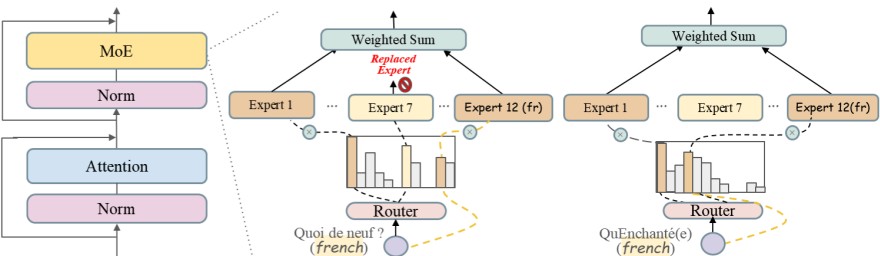

Figure 2: Hybrid routing ensures that the experts corresponding to the input token language are activated. As illustrated, if the weight of the language-specific expert do not rank among the top two, it will replace the expert with lower weights; otherwise, no changes will be made.

We propose new MoE consists of *language-specific experts* and *hybrid routing*, enhancing both language-specific expertise and transfer of general medical knowledge across languages.

Table 2: Comparison between Dense models and MoE models with various routing strategies.

| Method | Param. (B) Active | Total | Avg. Accuracy High | Low | Ar | De | En | Es | Fr | Hi | It | Ja | Ko | Pt | Ru | Zh |
|---|---|---|---|---|---|---|---|---|---|---|---|---|---|---|---|---|
| Dense Models before and after Training with Various Param. | | | | | | | | | | | | | | | | |
| Qwen-0.5B | 0.49 | 0.49 | 29.7 | 31.5 | 27.3 | 27.4 | 39.3 | 32.9 | 21.3 | 25.7 | 20.7 | 24.0 | 19.2 | 26.3 | 46.9 | 45.4 |
| *after training* | 0.49 | 0.49 | 37.8 | 24.6 | 34.8 | 33.7 | 46.7 | 40.5 | 33.0 | 31.0 | 31.5 | 31.5 | 35.5 | 30.8 | 53.8 | 51.0 |
| Same Active | **0.81** | 0.81 | 38.4 | 26.2 | 34.4 | 35.8 | 46.1 | 40.1 | 34.0 | 32.2 | 32.2 | 32.0 | 35.1 | 31.0 | 54.5 | 51.8 |
| Same Total | 3.95 | **3.95** | **42.0** | 30.9 | 36.4 | 40.8 | 47.8 | 42.1 | 38.0 | 34.6 | 43.2 | 32.4 | 38.7 | 31.9 | 62.9 | 54.8 |
| MoE Models Trained with Different Routing Strategies | | | | | | | | | | | | | | | | |
| Lang-Spec. | 0.81 | 3.95 | 30.9 | 26.1 | 28.8 | 28.6 | 39.1 | 33.3 | 16.5 | 24.5 | 21.4 | 27.8 | 31.0 | 52.3 | 42.8 | 53.0 |
| Top-$k$ | 0.81 | 3.95 | 39.7 | 29.9 | 34.5 | 36.9 | 43.7 | 38.9 | 39.9 | 32.2 | 37.7 | 30.5 | 35.9 | 34.8 | 58.2 | 53.2 |
| Hybrid-$k$ | **0.81** | **3.95** | 40.0 | **32.0** | 35.1 | 37.2 | 44.1 | 40.8 | 40.7 | 28.7 | 38.9 | 32.4 | 36.2 | 34.3 | 58.8 | 53.6 |

**Language-specific Experts** Inspired by Li et al. (2023c); Pfeiffer et al. (2022); Kwon & Chung (2023), we leverage the MoE structure to modularize the language-specific parameters in the medical domain. Specifically, we design language-specific experts to more effectively handle language-dependent knowledge and inputs. However, this has notable drawbacks: the knowledge encapsulated within each expert tends to be isolated, which can impede the learning of general medical knowledge.

**Hybrid Routing** To address this limitation and enhance general knowledge acquisition across languages, we propose cross-lingual routing within the MoE. This allows routing to go beyond language-specific experts, enabling knowledge to propagate across languages. As shown in Fig. 2, the result is a hybrid mechanism, where routing can target both language-specific Experts and dynamically route to other language experts; the latter is related to the input text itself.

In Hybrid-$k$, tokens can be routed not only to language-specific experts but also to cross-lingual experts. The *rationale* for cross-lingual routing is to view text as a tool for thought, capable of being expressed through various languages. Given the exceptional multilingual processing abilities of LLMs, it can be inferred that they adeptly switch between and intertwine multiple languages during text comprehension.

### 3.1.2 EXPERIMENTAL COMPARISONS BETWEEN ROUTING STRATEGIES

To validate the effectiveness of our method, we compared it with different routing strategies, specifically vanilla Top-$k$ and Language-Specific (*Lang-Spec.*) routing. *Lang-Spec.* routing refers to selecting the corresponding expert based on the input language type, while keeping a shared expert constantly activated to enhance performance and align the model's inference parameters.

**Experiment Settings** Considering its broad parameter base and impressive multilingual capabilities, we selected the Qwen2 series for our experiments. Specifically, we conducted experiments using the Qwen2-0.5B model, based on the training and evaluation datasets described in Sec. 2. To accurately construct a dense model of equivalent size with corresponding MoE model as baseline, we replicated the MLP following the approach used in MoE Upcycling and initialized the routing with an average distribution. Unlike MoE, the initialized dense model employs full activation instead of sparse activation. For **token language classification**, tokens are uniformly classified based on their source per document. To construct the **evaluation set for low-resource languages**, we evenly select 38 languages based on their geographical distribution. Similar to Sec. 2, we use Google Translate to translate the questions and answers from the medical-clinical section of the MMLU dataset, types of low-resource languages are detailed in the App. A.5. For Lang-Spec. routing, we use 12 experts plus one shared expert. For Hybrid-$k$ and Top-$k$ routing, 12 experts are used per layer. The activation count is fixed at two experts across all configurations ($k = 2$) if not specified. Additional training settings are provided in App. B.

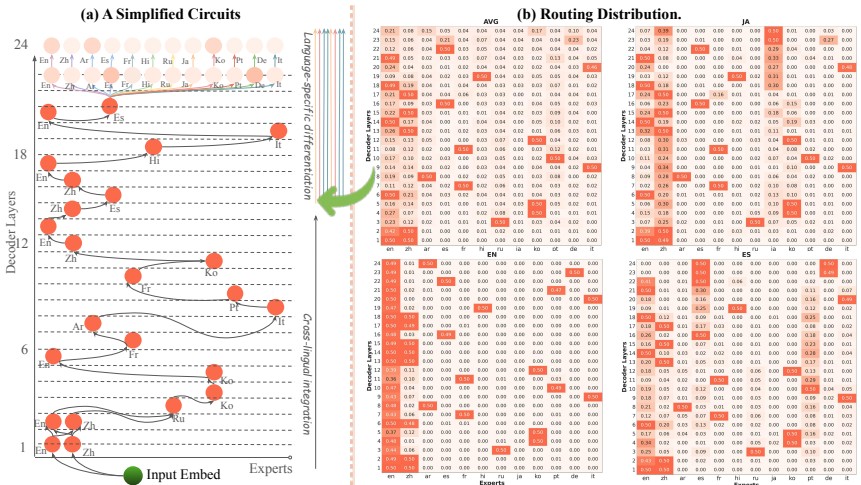

Figure 3: Visualization for routing patterns. `Right`: Hybrid-$k$ routing distribution. The x-axis represents language experts, with values indicating the proportion of tokens allocated to each expert. **AVG** denotes the aggregated routing distribution across 12 languages. `Left`: Visualization of **AVG** routing distribution from the perspective of *Information Flow Circuits*. We retained expert nodes with a token ratio of 0.5.

**Results**  Tab. 2 shows that the **base model** in high-resource languages is improved significantly after fine-tuning, but its performance in low-resource languages declines substantially. This indicates that partial language fine-tuning of dense models notably impacts their generalization capabilities to other languages. The **Language-Specific** routing of MoE provides interpretability but results in poor multilingual performance. In contrast, the **Top-$k$** MoE model demonstrates superior multilingual capability and generalization after fine-tuning compared to the trained base model, highlighting the effectiveness of MoE models over dense models. Moreover, **Hybrid-$k$** routing exhibits a clear advantages than **Top-$k$** in nearly all languages, especially in low-resource languages; this evidences its superior generalization.

## 3.2 INTERPRETABLE ROUTING ANALYSIS: INFORMATION FLOW CIRCUIT

To better interpret the routing patterns in multilingual context, we propose to formulate the routing pattern across layers as circuit in Sec. 3.2.1 and conduct some visualized study in Sec. 3.2.2.

### 3.2.1 FORMULATION OF INFORMATION FLOW IN ROUTING

During the routing process, each token is routed to both its language-specific expert and other language experts, enabling cross-lingual routing. This section aims to analyze the cross-lingual routing patterns for each language. Specifically, we seek to understand how tokens in a given language benefit from other language experts as they traverse from lower to higher layers. In other words, beyond its own language, the key question is: *How does input in each language leverage other language experts across different layers?* As mentioned in Sec. 3.1.1, the involvement of intermediate language experts may provide insights into the languages the model utilizes for internal thinking and reasoning.

To investigate the process, we propose a circuit-based formulation as below:

**Definition 1.** *Information Flow Circuit The sequence of Experts that each token passes from shallow to deeper abstractions forms a directed acyclic graph (DAG), which we refer to as a 'circuit'.*

To examine expert routing within the Hybrid-$k$ routing, we conduct an experiment to track token routing. We extracted varying amounts of data from 12 languages (details on data quantity and format are provided in App. D) to obtain approximately 80,000 tokens per language. These 12 single-language datasets were used as probing sets to record expert routing at the token level.

### 3.2.2 RESULTS

The visualization of the routing pattern is shown in Fig. 3. From the perspective of **single-language routing**, the routing results for Japanese and Spanish in the figure (routing distributions for other languages are detailed in the App.G) indicate that Chinese has a significant influence on Japanese, while Portuguese and English are also integrated into the information flow of Spanish. This phenomenon aligns well with linguistic priors: the linguistic development of Japanese has been influenced by Chinese, while Spanish, Portuguese, and English, all belonging to the Romance language family, have historically exerted mutual influence. From the perspective of cross-linguistic **routing for all 12 languages**, through the utilization of routing distribution analysis and *Information Flow Circuits* visualization, we observe that the information flow circuits exhibit cross-linguistic concentration in the early layers, while differentiation based on language occurs in the later layers. We refer to this phenomenon as "Spread Out in the End":

**Phenomenon 1.** *In the early layers, the model exhibits shared routing patterns across multiple languages. However, in the later layers, the model diverges, with tokens being routed to language-specific experts, allowing late routing to specialize in its respective language.*

### 3.3 A MoE VARIANT INSPIRED BY THE 'SPREAD OUT IN THE END' PHENOMENON: POST-MoE

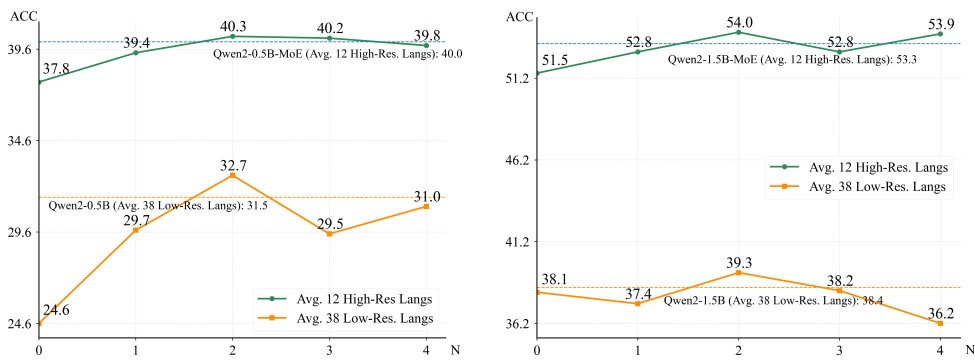

(a) Post-MoE from Qwen2-0.5B in Last N Layers   (b) Post-MoE from Qwen2-1.5B in Last N Layers

Figure 4: Analysis of Upcycling Layer Depths for the PostMoE Architecture. The X-axis represents the number of Upcycling layers applied in the final N layers, while the Y-axis indicates the model performance on both high- and low-resource languages. N=0 signifies direct fine-tuning of the model. Qwen2-0.5B-MoE and Qwen2-1.5B-MoE refer to standard MoE architectures trained with Hybrid routing.

Inspired by the phenomenon of "*Spread Out in the End*," we propose the Post-MoE architecture, which applies the Mixture of Experts (MoE) structure only in the final layers. Using this architecture, we further validate the observed phenomenon and investigate the impact of the number of MoE layers in the model's final layer on its performance.

**Experiment Settings** To validate the effectiveness of the multilingual mechanism and the Post-MoE architecture, we use Qwen2-0.5B-MoE and Qwen2-1.5B-MoE, along with the original base model as a baseline. Qwen2-0.5B-MoE and Qwen2-1.5B-MoE are standard MoE architectures trained with Hybrid routing. To

investigate the impact of the number of MoE layers in the final layer of the model on performance, we extend the MoE architecture in the last 1, 2, 3, and 4 layers using the hybrid routing method. We also evaluate the multilingual generalization capability of this architecture using the assessment set of 38 low-resource languages mentioned in Sec. 4.

**Results** As shown in Fig. 4, the architecture with MoE extended in the last two layers achieves the best performance, balancing both accuracy and multilingual generalization. The experimental results align closely with the routing pattern visualization in Fig. 3, further validating the "Spread Out in the End" phenomenon. In the next section, we will further apply this method to 50 languages to fully leverage the advantages and scalability of this architecture.

## 4 FURTHER SCALING TO 50 LANGUAGES

To further demonstrate the multilingual capabilities of the Post-MoE architecture, we selected 38 low-resource languages, expanding the language variety to 50 (shown in Tab. 3).

Table 3: Classification of Languages with Color Coding Based on Characteristics.

| Language Family | Languages |
| --- | --- |
| Sino-Tibetan | Chinese (Zh) |
| Altaic | Korean (Ko), Japanese (Ja), Mongolian (Ne) |
| Australasian | Thai (Th), Vietnamese (Vi), Laotian (Lo) |
| Austronesian | Malagasy (Mg), Cebuano (Ceb), Sundanese (Su), Ilokano (Ilo), Dogue (Doi) |
| Indo-European | English (En), German (De), Portuguese (Pt), Spanish (Es), French (Fr), Russian (Ru) Italian (It), Croatian (Hr), Galician (Gl), Czech (Cs), Corsican (Co), Latin (La), Ukrainian (Uk), Bosnian (Bs), Bulgarian (Bg), Esperanto (Eo), Maithili (Mai), Albanian (Sq), Danish (Da), Sanskrit (Sa), Norwegian (No), Guarani (Gn), Serbian (Sr), Slovak (Sk), Scottish Gaelic (Gd), Luxembourgish (Lb), Hindi (Hi) |
| Afro-Asian | Arabic (Ar), Kurdish (Sorani) (Ckb), Maltese (Mt), Hebrew (He) |
| Kongolese | Lingala (Ln), Bambara (Bm), Swahili (Sw), Sepeti (Nso), Igbo (Ig), Kinyarwanda (Rw), Hausa (Ha) |

### 4.1 MIXTURE OF LANGUAGE FAMILY EXPERTS

Following the Hybrid-$k$ which adopts language-specifc experts, training LLMs with $n$ languages would require $n$ language-specific experts. This expansion strategy would lead to an substantial growth in model parameters, making the application impractical. To address this challenge, we propose adapting Post-MoE by introducing the Mixture of Language Family Experts, which groups the 50 languages into 7 established linguistic families, as detailed in Tab. 3. This approach named '*Apollo-MoE*' ensures that scaling to additional languages does not necessitate a corresponding increase in parameters. Each language family employs hybrid routing tailored to its linguistic characteristics, enabling more efficient training. This method might facilitate scalable and robust multilingual performance across a wide range of languages. After training 50 languages on Dense models and ablation study of various routing strategies in PostMoE, we further verify the effective multilingual generalization of *Apollo-MoE* in App.E.

### 4.2 EXPERIMENTS

**Experiment Settings** Given the extremely limited data for these rare languages, we extracted 2,000 samples from English data and used Google Translate to create **training corpora** for each language. The clinical-knowledge section of MMLU was translated to serve as the corresponding **evaluation set**. We trained the Post-MoE model, named Apollo-MoE, using Qwen2-0.5B, 1.5B, and 7B as base models with a Mixture of Language Family Experts. We evaluated Apollo-MoE on benchmarks for high-resource (see App. 7) and low-resource languages and compared it with close-source and high-performing open-source medical models.

Table 4: Main Results on 50 languages comparing to existing LLMs

| Model | Active Param. | Average Acc. High | Low | Ar | De | En | Es | Fr | Hi | It | Ja | Ko | Pt | Ru | Zh |
|---|---|---|---|---|---|---|---|---|---|---|---|---|---|---|---|
| **Closed-source Models** | | | | | | | | | | | | | | | |
| Gemini 1.5 Pro | - | 74.6 | 73.6 | 74.9 | 80.4 | 63.2 | 81.9 | 87.5 | 75.4 | 69.1 | 68.9 | 78.6 | 79.9 | 64.8 | 70.0 |
| GPT-4 Turbo | - | 79.4 | 73.3 | 79.8 | 84.7 | 68.5 | 85.8 | 89.7 | 72.8 | 80.9 | 77.5 | 79.5 | 87.0 | 78.1 | 68.9 |
| Claude 3 Opus | - | 81.9 | 76.5 | 78.8 | 87.2 | 69.8 | 87.9 | 90.7 | 78.6 | 84.6 | 82.1 | 85.2 | 86.7 | 80.1 | 70.9 |
| GPT-4o | - | **85.7** | **81.1** | 80.9 | 90.5 | 70.5 | 90.7 | 93.5 | 86.9 | 85.1 | 89.5 | 90.2 | 90.8 | 78.5 | 80.9 |
| GPT-4o mini | - | 77.6 | 69.8 | 74.3 | 80.4 | 70.2 | 82.3 | 84.5 | 75.4 | 77.6 | 76.2 | 77.1 | 82.3 | 80.9 | 67.5 |
| **Open-source Models** | | | | | | | | | | | | | | | |
| JSL-MedPhi2 | 2.78 B | 29.0 | 30.7 | 25.0 | 30.8 | 42.1 | 34.6 | 30.6 | 13.2 | 26.6 | 17.9 | 21.3 | 29.7 | 48.8 | 26.8 |
| MMed-Llama-3 | 8.03 B | 40.2 | 36.5 | 29.5 | 40.3 | 54.7 | 63.6 | 62.0 | 38.8 | 20.7 | 47.3 | 20.3 | 25.9 | 62.5 | 16.7 |
| OpenBioLLM | 8.03 B | 46.1 | 34.7 | 27.5 | 58.1 | 49.6 | 59.2 | 53.3 | 44.4 | 41.5 | 39.1 | 46.7 | 27.0 | 64.8 | 41.8 |
| Llama3 | 8.03 B | 49.3 | 33.3 | 33.5 | 55.9 | 48.4 | 60.5 | 58.4 | 48.0 | 39.9 | 38.9 | 49.5 | 51.5 | 63.3 | 43.6 |
| **Our Models** | | | | | | | | | | | | | | | |
| Apollo-MoE | 0.52 B | 40.5 | 34.6 | 36.3 | 38.2 | 45.4 | 39.8 | 38.4 | 33.1 | 39.9 | 26.9 | 35.2 | 37.3 | 64.1 | 51.3 |
| Apollo-MoE | 1.63 B | 54.8 | 44.9 | 47.2 | 53.8 | 56.5 | 52.5 | 53.3 | 39.5 | 54.4 | 45.7 | 49.5 | 57.3 | 69.1 | 66.8 |
| Apollo-MoE | 8.02 B | **69.9** | **58.3** | 58.3 | 73.5 | 73.1 | 69.4 | 72.4 | 56.9 | 71.9 | 62.4 | 68.4 | 73.8 | 74.2 | 84.1 |

**Results** Tab. 4 demonstrate that our Apollo-MoE models outperform other models of similar sizes in both high- and low-resource languages. The 2B model (1.63B active) achieves 54.8 in high-resource and 44.9 in low-resource languages, surpassing open-source models with 8B parameters. The 10B model (8.02B active) leads all open-source models, achieving 69.9 in high-resource and 58.3 in low-resource languages. Notably, Apollo-MoE excels in high-resource languages like English, French, and Spanish, with particularly strong performance in French, and outperforms other models significantly in low-resource languages like Arabic and Hindi.

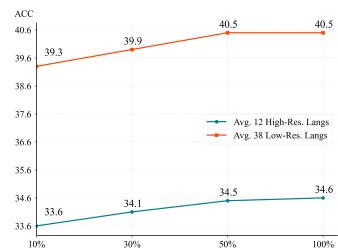

Figure 5: Data Scale Performance.

**Ablation Study** Tab. 5 provides a detailed comparison of routing performance across 50 languages, with two experts routed among 7 linguistic family experts in the Post-MoE model (k=2). It shows that routing strategies consistently outperform Dense models, with Hybrid-$k$ routing achieving an average high-resource language accuracy of 54.8 for Qwen2-1.5B, compared to 52.2 for Dense. Additionally, Hybrid-$k$ routing shows a clear advantage over Top-$k$ routing, particularly in low-resource languages, where the Qwen2-7B model achieves an average accuracy of 58.3 with Hybrid-$k$, compared to 56.7 with Top-$k$.

**On the Data Scale** To investigate the sensitivity to data scale, we trained Post-MoE model with Qwen2-0.5B base using a combination of high-resource languages along with 10%, 30%, 50%, and 100% of the data from low-resource languages, corresponding to 200, 600, 1,000, and 2,000 data examples per language, respectively. As shown in Fig. 5, the model's performance across various languages improves with increasing data but eventually plateaus. This indicates that the proposed method is not heavily data-dependent, reaching saturation with as few as 2,000 data examples; this is especially beneficial for low-resources languages.

## 5 RELATED WORK

**Mixture-of-Experts and Sparse Upcycling** Mixture-of-Experts (MoE) models differ from traditional dense models by activating only a subset of parameters, or "experts", for each input token. This selective activation reduces computational costs while preserving model capacity. Introduced by Shazeer et al. (2017), MoEs have evolved in models like GShard (Lepikhin et al., 2020), Switch Transformers (Fedus et al., 2022), and

Table 5: Ablation Study between Dense and PostMoE Models across 50 Languages at Various Scales.

| Languages | Active Param. | Average acc. High | Low | Ha | Sr | La | Gn | Doi | Da | Ln | Ceb | Mai | Mg | Ilo |
|---|---|---|---|---|---|---|---|---|---|---|---|---|---|---|
| Qwen2-0.5B | 0.49 B | 29.7 | 31.5 | 31.1 | 30.7 | 36.4 | 28.4 | 27.7 | 33.3 | 29.9 | 34.5 | 27.3 | 34.8 | 30.7 |
| Dense | 0.49 B | 39.2 | 33.2 | 33.7 | 25.0 | 30.7 | 33.3 | 32.6 | 36.0 | 34.5 | 36.7 | 31.8 | 34.1 | 35.2 |
| Dense Same Active | 0.52 B | 39.4 | 34.0 | 34.1 | 25.5 | 32.7 | 34.2 | 34.2 | 34.4 | 38.9 | 39.2 | 32.5 | 32.2 | 35.9 |
| Top-$k$ routing | 0.52 B | 39.0 | 33.4 | 26.9 | 17.8 | 37.5 | 34.1 | 29.9 | 36.7 | 33.7 | 32.2 | 28.8 | 32.2 | 32.6 |
| Hybrid-$k$ routing | 0.52 B | **40.5** | **34.6** | 31.1 | 25.4 | 38.3 | 34.8 | 36.0 | 39.8 | 41.3 | 40.9 | 33.7 | 37.9 | 39.8 |
| Qwen2-1.5B | 1.54 B | 42.9 | 38.4 | 36.0 | 44.7 | 39.0 | 37.5 | 31.8 | 45.1 | 34.8 | 42.4 | 34.1 | 39.0 | 35.6 |
| Dense | 1.54 B | 52.2 | 43.7 | 36.7 | 30.3 | 51.1 | 47.0 | 37.1 | 51.1 | 44.3 | 48.9 | 39.8 | 40.5 | 45.8 |
| Dense Same Active | 1.63 B | 52.8 | 44.1 | 38.8 | 28.3 | 50.0 | 49.1 | 39.9 | 53.1 | 44.7 | 48.1 | 40.5 | 40.2 | 46.5 |
| Top-$k$ routing | 1.63 B | 53.6 | 42.6 | 39.8 | 24.6 | 47.7 | 44.7 | 36.0 | 50.4 | 34.5 | 48.5 | 35.6 | 41.7 | 43.9 |
| Hybrid-$k$ routing | 1.63 B | **54.8** | **44.9** | 39.0 | 33.7 | 48.5 | 45.8 | 41.3 | 55.3 | 43.9 | 50.8 | 42.8 | 41.7 | 46.2 |
| Qwen2-7B | 7.62 B | 55.2 | 49.2 | 34.1 | 57.6 | 52.3 | 43.2 | 40.9 | 63.3 | 38.3 | 58.3 | 48.5 | 37.1 | 45.5 |
| Dense | 7.62 B | 69.0 | 55.7 | 37.9 | 31.1 | 60.6 | 54.9 | 55.7 | 68.6 | 47.3 | 64.4 | 55.7 | 44.7 | 64.4 |
| Dense Same Active | 8.02 B | 68.5 | 56.3 | 42.2 | 39.7 | 59.5 | 57.9 | 55.5 | 69.3 | 51.1 | 65.9 | 57.8 | 46.8 | 61.7 |
| Top-$k$ Routing | 8.02 B | 68.6 | 56.7 | 37.1 | 39.4 | 60.6 | 56.1 | 50.8 | 67.8 | 53.4 | 67.0 | 59.8 | 46.2 | 61.0 |
| Hybrid-$k$ Routing | 8.02 B | **69.9** | **58.3** | 48.5 | 42.0 | 62.9 | 61.0 | 54.5 | 71.2 | 53.0 | 68.6 | 58.3 | 50.8 | 64.4 |

Mixtral (Jiang et al., 2024), showing improved performance in large-scale tasks. Recent advancements like Sparse Upcycling (Komatsuzaki et al., 2022) efficiently initialize MoEs by replicating dense model parameters. BTX (Sukhbaatar et al., 2024) further refines MoE efficiency by upcycling specialized models.

**Multilingual Medical LLMs** Recent research has focused on expanding multilingual medical data and creating models for linguistically diverse populations. A LLM integrated with machine translation has been proposed to address the scarcity of such datasets (Gangavarapu, 2024). Furthermore, multilingual medical corpora and benchmarks across six languages have been developed (Qiu et al., 2024; Wang et al., 2024). Our work builds on these efforts by investigating multilingual patterns and expanding it to 50 languages.

**Multilingual Capability Enhancement** Research on Multilingual Models has primarily aimed to enhance multilingual capabilities and understand their mechanisms. GreenPLM (Zeng et al., 2023) shares the same motivation with our work to efficiently expand the model's multilingual capabilities. Efforts have improved performance through translation (Liang et al., 2023) and cross-lingual alignment (Salesky et al., 2023). Techniques like cross-lingual transfer (Kim et al., 2017) and continuous training in specific languages have further advanced LLMs (Cui et al., 2023), while training from scratch shows potential (Muennighoff et al., 2023). Recent models (Zhao et al., 2023; Nguyen et al., 2024) also exhibit strong multilingual abilities without explicit language alignment. Tang et al. (2024) and Zhao et al. (2024) employ neuron analysis method (Mu & Andreas, 2020) to investigate the mechanism, but they typically focus on fewer than seven languages.

## 6 CONCLUSION

In conclusion, this work advances the development of multilingual medical LLMs by addressing key challenges in data scarcity and model scalability. We first construct a high-quality medical dataset covering 12 high-resource languages. Subsequently, we propose Hybrid-$k$ routing to explore multilingual training in a modular manner in MoE. Based on the Hybrid-$k$ routing, we propose a circuits-based paradigm for interpreting information flow in multilingual contexts. The circuit analysis reveal the "Spread Out in the End" mechanism, based on which we introduced the Post-MoE architecture and demonstrate its superior multilingual capabilities. Furthermore, by introducing language family experts, we efficiently extend Post-MoE's capabilities to 50 languages with limited data, demonstrating scalability without additional parameters.

LIMITATIONS

Post-MoE expansion in the last two layers has shown good results. However, our base model for Post-MoE expansion only goes up to 7B parameters. For larger models, the optimal effect might not be achieved by expanding just the last two layers; a proportional calculation of layers may be required. Nonetheless, using the last two layers already yields satisfactory results for the 7B model. Research on MoE at the module level is still in its early stages. Although Post-MoE demonstrates superior performance across multiple languages, there is still significant room for improvement in model performance.

**Future Work**  In practical applications, the Mixture of Language Family Experts in Post-MoE enables versatile task adaptation and efficient scalability. Additionally, by treating modalities as distinct languages, the proposed framework facilitates studying multimodal interaction dynamics. Theoretically, while many studies have highlighted the presence of multilingual neurons at the bottom layers of the model, in circuit-based paradigm we further observed that activation parameters at bottom layers are concentrated in the Chinese and English language modules rather than being distributed across various language modules and hypothesize that this may reflect a model strategy that relies on core language experts at the bottom layers for multilingual translation. The observation provides an additional dimension (language experts) for analyzing multilingual mechanisms beyond the depth perspective. It prompts further exploration into the selection of language experts at specific depths and their inter-layer interactions.

ACKNOWLEDGEMENT

This work was supported by the Shenzhen Science and Technology Program (JCYJ20220818103001002), Shenzhen Doctoral Startup Funding (RCBS20221008093330065), Tianyuan Fund for Mathematics of National Natural Science Foundation of China (NSFC) (12326608), Shenzhen Key Laboratory of Cross-Modal Cognitive Computing (grant number ZDSYS20230626091302006), and Shenzhen Stability Science Program 2023.

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

# A    DETAILS OF DATASETS, BENCHMARK

## A.1    DATASET COLLECTION

| Language | Source |
|---|---|
| **Books** | |
| English | Pile Dataset (Gao et al., 2020) |
| Chinese | MedQA (Jin et al., 2020) |
| Russian | ru-medical-textbooks [Link] |
| German | de-books [Link] |
| Korean | ko-books [Link] |
| **Papers** | |
| English | PubMed (Roberts, 2001) |
| Chinese | Paper from Chinese Medical Association [Link] |
| French | MORFITT (Labrak et al., 2023c), CLEAR (Grabar & Cardon, 2018) |
| Spanish | Mesinesp (Gasco et al., 2021) |
| Russian | ru-medical-paper [Link] |
| German | Germany part of Multilingual Medical Corpora [Link] |
| **Encyclopedias** | |
| ⟨Multiple⟩ | ⟨English, Russian, Hindi, Arabic, German, Italian, Korean, Japanese⟩wiki [Link] |
| French | CLEAR (Grabar & Cardon, 2018) |
| Hindi | HHD corpus (Jain & Arora, 2018) |
| Portuguese | pt-medical-wiki [Link] |
| **Dialogues** | |
| Chinese | HuatuoGPT dataset (Zhang et al., 2023), Huotuo_26M (Li et al., 2023b) |
| English | PMC-Patients (Zhao et al., 2022) |
| Arabic | MAQA (Abdelhay & Mohammed, 2022) |
| Russian | RuMedPrimeData [Link] |
| Portuguese | askD [Link] |
| Italian | MedQuaAD-Italian [Link] |
| Korean | MedGPT-5k-ko [Link], ko-medical-chat [Link] |
| Japanese | Real-MedNLP Test Collection [Link], ja-medial-progress-notes [Link] |
| **Exam** | |
| Chinese | CMB (Wang et al., 2023), CMExam (Liu et al., 2024), MedQA (Zhang et al., 2018) |
| English | MedQA, Medmcqa (Pal et al., 2022), Pubmedqa (Jin et al., 2019) |
| Spanish | HEAD-QA (Vilares & Gómez-Rodríguez, 2019) |
| French | Frenchmcqa (Labrak et al., 2023a) |
| Italian | MedExpQA (Alonso et al., 2024a) |
| Russian | RuMedBench (Blinov et al., 2022a), BioInstructQA (Labrak et al., 2024a) |
| Japanese | IgakuQA (Kasai et al., 2023b) |
| Korean | KorMedMCQA (Kweon et al., 2024) |
| German | BioInstructQA |
| **Guideline** | |
| English | NICE [Link], PubMed, SPOR [Link] |
| ⟨Multiple⟩ | ⟨Spanish, German, French⟩MSD-instruct [Link] |
| Korean | Korean-guidelines-for-primary-physicians [Link] |
| **Web** | |
| Chinese | WUdao Dataset [Link] |
| English | C4 Dataset (Raffel et al., 2019) |
| Spanish | CoWeSe (Carrino et al., 2021), medical-eval-pt [Link] |
| Italian | it-medical-corpus [Link], it-biomedical-dataset [Link] |
| German | opus-medical-de-en [Link] |
| Russian | medical-qa-ru-data [Link] |
| Japanese | MedNLP-SC Social Media Corpus [Link] |
| **General** | |
| ⟨Multiple⟩ | ⟨French, Spanish, Arabic, Hindi, German, Russian, Italian, Portuguese, Japanese, Korean⟩Alpaca [Link], Sharegpt [Link] |
| Ch,En | Alpaca, Sharegpt, WizardLM Dataset |
| English | belebele benchmark (Bandarkar et al., 2023), ai2_arc (Clark et al., 2018), Capybara (Daniele & Suphavadeeprasit, 2023) |
| **Math** | |
| ⟨Multiple⟩ | MathInstruct (Yue et al., 2023) |
| **Code** | |
| English | Python-Alpaca [Link] |
| Chinese | Leetcode-ZH-11k [Link] |

Table 6: The detailed sources of training dataset.

## A.2 DATASET PROCESSION

Adhering to the established data processing standards, we segmented sentences into chunks, filtered them for medical relevance, and reformatted them into a QA format with prompt shown in App. C.2. The processing details are as follows:

**Books**  For English books, we use medical dictionary[3] to filter the Pile Dataset and select books where medical terms account for more than 4% of the total words, resulting in 2,312 medical-related books. For Chinese books, we follow MedQA to collect medical textbooks included in the five-year and eight-year medical student training programs in mainland China, finally obtaining 90 books. For Russian books, we use Russian medical textbooks as the data source. For German medical book data, we first divided the German book data into multiple blocks with a maximum of 512 characters, then filtered the data using 2,590 highly relevant German medical terms[4], and finally rewrote it into QA form. Regarding Korean medical book data, we adopt the same method and Korean medical dictionary[5] to filter out the medical data from Korean books.

**Papers**  For English papers, we sample the public data in PubMed and obtain 878,241 medical abstracts. For Chinese papers, we also screen a total of 177,261 abstracts of papers published by the Chinese Medical Association. For French papers, we use the MORFITT dataset and the scientific article portion of the CLEAR. For the Spanish paper, we use paper abstracts open sourced by the Mesinesp. For medical literature in German and Russian, we directly divide the medical data into blocks of 512 and rewrite them into QA format for use respectively.

**Encyclopedias**  For the English Encyclopedia, we also use the English Medical Dictionary to filter out 36107 medical-related wiki pages from wiki dataset. For the French encyclopedia, we select the encyclopedia articles part of the CLEAR and filtered[6] wiki data for supplementation. For the Hindi encyclopedia, we choose the HHD corpus, which crawls descriptions of people, diseases, medical consumer products, and symptoms from Indian websites. For Russian, Hindi, Arabic, German, Italian, and Korean, we filtered wiki data using medical dictionaries of corresponding languages[7]. For Portuguese, we directly used the pt-medical-wiki data. As for Japanese data, we screened wiki using medical dictionaries[8][9] to obtain Japanese medical encyclopedia data.

**Doctor-Patient Dialogues**  For Chinese, we directly use the HuatuoGPT dataset and the simplified data set in Huatuo_26M. For English, we construct a multi-turn conversation data set based on PMC-Patients using ChatGPT with the prompt shown in Fig.7. For Arabic, we extract high-quality questions and answers with both question and answer lengths greater than 128 from the largest Arabic healthcare question and answer dataset MAQA. For Russian, we use RuMedPrimeData from outpatient hospital patients. For Portuguese, we utilize medical Q&A data from askD. For Italian, we employ doctor-patient data from MedQuAD-Italian. For Korean, we filter the MedGPT-5k-ko and ko-medical-chat data for medically rich doctor-patient dialogues with the dictionary. For Japanese, we rewrite QA pairs from Real-MedNLP and medical progress note to serve as Japanese doctor-patient dialogues.

**Exams**  For the Chinese exam, we collect training sets of CMB, CMExam, and MedQA. For the English exam, we collect the training sets of MedQA, Medmcqa and Pubmedqa. For the Spanish and French exam, we select the training set of HEAD-QA and Frenchmcqa  separately. For Italian, we use the training portion defined by MedExpQA as the Italian medical exam data. For Russian, we utilize the training part from

---

[3]https://www.nlm.nih.gov/research/umls/new_users/online_learning/LEX_001.html
[4]https://medlineplus.gov/languages/russian.html
[5]https://medlineplus.gov/languages/korean.html
[6]https://medlineplus.gov/languages/french.html
[7]https://medlineplus.gov/languages/languages.html
[8]https://sociocom.naist.jp/manbyo-dic-en/
[9]https://sociocom.naist.jp/hyakuyaku-dic-en/

RuMedBench and the exam portion from BioiInstructQA. For Japanese, we divided IgakuQA into training and test parts at a 6:4 ratio, using them as training data for the Japanese medical exam and the Japanese medical benchmark, respectively. For Korean, we use the training part provided by KorMedMCQA. For German traning set, we employ German sections MedQA, PubMedQA, MedQA-5-options, and MedMCQA in BioInstructQA.

**Guidelines**   For English Guidelines, we select data from three sub-items of NICE, PubMed and SPOR in the clinical guidelines introduced by Meditron (Chen et al., 2023b). Regarding the guidelines for Spanish and German, we use data from MSD-instruct. Additionally, we incorporate the French sections as supplementary material for the French guidelines. For the Korean section, we extracted and segmented content in 1024 characters from Korean-guidelines-for-primary-physicians, using the medical dictionary for data filtering.

**General**   For all 12 languages, we use the translation and original data of Sharegpt and Alpaca. For Chinese, we additionally make use of WizardLM Dataset generated by GPT-4 based on WizardLM Method (Xu et al., 2023). For English, in addition to adding the WizardLM Dataset, we also add belebele to enhance multi-language reading comprehension capabilities, ai2_arc to enhance abstract reasoning capabilities, Capybara to enhance instruction following capabilities.

**Web**   For Chinese, we use the medical dictionary[10] to filter out medical-related articles from the Wudao Dataset. For English, we use the English Medical Vocabulary[11] to filter out medical related articles in C4 Datase. For Spanish, we sampled 10% of CoWeSe Dataset and used the medical-eval-pt data for supplementation. For Italian, we utilize data from it-medical-corpus and it-biomedical-dataset. For German, we use the German portion of data from opus-medical-de-en. For Russian, we employ data from medical-qa-ru-data, which contains 190,335 Russian Q&A posts from a medical-related forum. For Japanese, we use the medical web data from MedNLP-SC Social Media Corpus.

**Math**   For mathematical abilities, we choose MathInstruct, a composite dataset containing various mathematics-related tasks and problem formats.

**Code**   We choose Python-Alpaca and Leetcode-ZH-11k respectively to strengthen the ability to solve coding tasks in Chinese and English.

## A.3   BENCHMARK ACROSS 12 HIGH-RESOURCE LANGUAGES

To ensure the reliability of our evaluation, we use publicly available multilingual medical benchmarks consisting of multiple-choice medical questions, with accuracy as the evaluation metric. For languages with limited evaluations, following multilingual evaluation method of Llama3 (Dubey et al., 2024) we translate MMLU (Hendrycks et al., 2020) questions and answers using Google Translate. We employed a random selection of 3-shot queries to pose questions to the model, followed by answer extraction and evaluation of the model's responses. Tab. 7 shows the sources of the 12 multilingual benchmark and the test data samples are shown in Fig. 6.

Specifically, we follow Med-PaLM2 (Singhal et al., 2023) and select six subcategories in MMLU: Clinical knowledge, Medical genetics, Anatomy, Professional medicine, College biology, and College medicine. For MedQA, we choose the 4-options version. For CMMLU, we select seven subdirectories: Anatomy, Clinical knowledge, College medicine, Genetics, Nutrition, Traditional chinese medicine, and Virology.

---

**Examples**

**User:** You are a medical doctor answering real-world medical exam questions. Select one correct answer from A to D.
Question: Rickets of prematurity is associated with:
Options:
(A) Hypocalcaemic convulsions
(B) Use of frusemide diuretic
(C) Vitamin D deficiency in the mother
(D) All of the options given are correct
**Assistant:** The correct answer is (D).

**User:** You are a medical doctor answering real-world medical exam questions. Select one correct answer from A to D.
Question: Diagnosis of iron deficiency can be complicated by concurrent infection since many markers of iron status are altered by infection. Which of the following combinations of iron status markers is likely to be found in a person with both iron deficiency and a severe infection?
Options:
(A) Low haemoglobin, high ferritin, high serum transferrin receptors, high hepcidin
(B) Low haemoglobin, low ferritin, high serum transferrin receptors, low hepcidin
(C) Low haemoglobin, low ferritin, normal serum transferrin receptors, high hepcidin
(D) Low haemoglobin, low ferritin, low serum transferrin receptors, high hepcidin
**Assistant:** The correct answer is (A).

**User:** You are a medical doctor answering real-world medical exam questions. Select one correct answer from A to D.
Question: What three factors regulate stroke volume?
Options:
(A) Blood volume, preload, and afterload.
(B) Preload, contractility, and afterload.
(C) Contractility, blood volume, and blood pressure.
(D) Cardiac output, contractility, and blood volume.
**Assistant:** The correct answer is (B).

**User:** You are a medical doctor answering real-world medical exam questions. Select one correct answer from A to D.
Question: A lesion causing compression of the facial nerve at the stylomastoid foramen will cause ipsilateral
Options:
(A) paralysis of the facial muscles.
(B) paralysis of the facial muscles and loss of taste.
(C) paralysis of the facial muscles, loss of taste and lacrimation.
(D) paralysis of the facial muscles, loss of taste, lacrimation and decreased salivation.
**Assistant:**

Figure 6: Sample English Evaluation Data (Similar to Other Languages).

Table 7: Benchmark across 12 Languages.

| Language | Benchmark Composition |
|---|---|
| Chinese | MedQA-MCMLE (Zhang et al., 2018), Medical parts of CMMLU (Li et al., 2023a) |
| English | MedQA-USMLE (Zhang et al., 2018), MedMCQA (Pal et al., 2022) |
| | Medical parts of MMLU (Hendrycks et al., 2020) |
| Spanish | HEAD-QA (Vilares & Gómez-Rodríguez, 2019) |
| Russian | RuMedBench (Blinov et al., 2022b) |
| Korean | KorMedMCQA (Kweon et al., 2024) |
| Japanese | IgakuQA (Kasai et al., 2023a) |
| German | MMLU part of BioInstructQA (Labrak et al., 2024b) |
| Portuguese | MMLU part of BioInstructQA (Labrak et al., 2024b) |
| Italy | MedExpQA (Alonso et al., 2024b), Translated medical parts of MMLU |
| Arabic | Translated medical parts of MMLU |
| Hindi | Translated medical parts of MMLU |
| French | FrenchMedMCQA (Labrak et al., 2023b), Translated medical part of MMLU |

Table 8: Performance of Diverse Models across 12 Languages, and the comparison before and after training of base models.

| Model | Size | Ar | De | En | Es | Fr | Hi | It | Ja | Ko | Pt | Ru | Zh | Avg. |
|---|---|---|---|---|---|---|---|---|---|---|---|---|---|---|
| | | | | | | **Closed-source Models** | | | | | | | | |
| Gemini-1.5 Pro | - | 74.9 | 80.4 | 63.2 | 81.9 | 87.5 | 75.4 | 69.1 | 68.9 | 78.6 | 79.9 | 64.8 | 70.0 | 74.6 |
| GPT-4 Turbo | - | 79.8 | 84.7 | 68.5 | 85.8 | 89.7 | 72.8 | 80.9 | 77.5 | 79.5 | 87.0 | 78.1 | 68.9 | 79.4 |
| Claude-3 Opus | - | 78.8 | 87.2 | 69.8 | 87.9 | 90.7 | 78.6 | 84.6 | 82.1 | 85.2 | 86.7 | 80.1 | 70.9 | 81.9 |
| GPT-4o | - | 80.9 | 90.5 | 70.5 | 90.7 | 93.5 | 86.9 | 85.1 | 89.5 | 90.2 | 90.8 | 78.5 | 80.9 | **85.7** |
| GPT-4o mini | - | 74.1 | 80.7 | 70.5 | 82.3 | 84.7 | 75.2 | 77.7 | 76.4 | 77.2 | 82.2 | 80.9 | 67.9 | 78.2 |
| | | | | | | **Open-source Models** | | | | | | | | |
| MMed-Llama-3 | 8B | 29.5 | 40.3 | 54.7 | 63.6 | 62.0 | 38.8 | 20.7 | 47.3 | 20.3 | 25.9 | 62.5 | 16.7 | 40.2 |
| OpenBioLLM | 8B | 27.5 | 58.1 | 49.6 | 59.2 | 53.3 | 44.4 | 41.5 | 39.1 | 46.7 | 27.0 | 64.8 | 41.8 | 46.1 |
| Llama3 | 8B | 33.5 | 55.9 | 48.4 | 60.5 | 58.4 | 48.0 | 39.9 | 38.9 | 49.5 | 51.5 | 63.3 | 43.6 | 49.3 |
| | | | | | | **Base Models before and after Training** | | | | | | | | |
| Gemma2 | 2B | 34.6 | 39.5 | 47.7 | 46.7 | 37.1 | 37.7 | 32.4 | 30.3 | 34.2 | 38.3 | 62.5 | 36.1 | 39.8 |
| *Aft.* | 2B | 44.1 | 51.0 | 59.6 | 50.9 | 47.3 | 39.4 | 46.8 | 44.5 | 48.2 | 53.5 | 62.5 | 64.5 | 51.0 |
| Phi-3 | 3.8B | 18.3 | 41.0 | 43.4 | 40.0 | 40.8 | 19.3 | 34.6 | 17.0 | 18.1 | 37.8 | 57.4 | 22.6 | 32.5 |
| *Aft.* | 3.8B | 36.5 | 60.7 | 66.5 | 59.4 | 57.0 | 36.4 | 59.0 | 27.1 | 46.4 | 63.3 | 59.8 | 65.4 | 53.1 |
| Qwen2 | 7B | 45.7 | 54.0 | 58.6 | 60.5 | 58.6 | 35.4 | 44.7 | 45.4 | 52.6 | 48.0 | 77.3 | 81.5 | 55.2 |
| *Aft.* | 7B | 54.8 | 71.2 | 62.6 | 69.7 | 70.4 | 55.0 | 65.4 | 63.5 | 67.7 | 66.1 | 78.1 | 85.4 | 67.5 |
| Gemma2 | 9B | 60.7 | 69.9 | 59.1 | 71.0 | 71.3 | 63.2 | 70.9 | 58.0 | 66.3 | 72.6 | 71.5 | 56.5 | 65.9 |
| *Aft.* | 9B | 65.2 | 78.5 | 77.2 | 72.9 | 75.6 | 65.8 | 75.6 | 67.8 | 70.6 | 77.5 | 69.9 | 76.9 | 72.8 |

## A.4 MORE EXPERIMENTAL ANALYSIS ON THE DATASET

To further validate the quality and effectiveness of the data, we increase both model diversity and parameter count.

**Experiment Settings** For **Baselines**, We select Gemma2-2B (Team et al., 2024b), Phi-3-mini-4k (Abdin et al., 2024) and Qwen2-7B (Yang et al., 2024) as base models. We simultaneously selected the closed-source models, Claude3 Opus (Anthropic, 2024), and GPT-4o (Achiam et al., 2023), as well as highly competitive

---
[10]http://thuocl.thunlp.org/#yixue
[11]https://www.nlm.nih.gov/research/umls/new_users/online_learning/LEX_001.html

open-source models MMed-Llama-3-8B (Qiu et al., 2024), LLama3-OpenBioLLM-8B (Ankit Pal, 2024), and LLama3-8B (Dubey et al., 2024) as baselines. For **Training**, We use full-parameter fine-tuning with a learning rate of 1e-5 for the 7B and 9B models, and a learning rate of 1e-4 for smaller models. All model use a batch size of 32 and a sequence length of 4096 for training on 8x NVIDIA A800-SXM4-80GB GPUs.

**Result Analysis**   As shown in Tab. 8, the performance of each model significantly improved after training with 12 high-resource languages data, further validating the effectiveness of data quality. The fine-tuned models excelled in the medical domain, significantly surpassing open-source medical models with similar parameter counts. Additionally, the medical model fully fine-tuned based on Gemma2-9B performed nearly as well as large-scale closed-source models.

A.5    LOW-RESOURCE LANGUAGES DATASET&BENCHMARK

To assess the model's generalization capability in multilingual contexts, we selected 38 low-resource languages with 12 high-resource languages, totaling 50 languages.

- **High-Resource Languages** English, Chinese, German, Portuguese, Spanish, French, Russian, Hindi, Italian, Korean, Japanese, Arabic
- **Low-Resource Languages** Mongolian, Thai, Vietnamese, Lao, Malagasy, Cebuano, Sundanese, Ilokano, Dogue, Croatian, Galician, Czech, Corsican, Luxembourgish, Latin, Ukrainian, Bosnian, Bulgarian, Esperanto, Maithili, Serbian, Albanian, Slovak, Danish, Sanskrit, Norwegian, Guarani, Scottish Gaelic, Kurdish (Sorani), Maltese, Hebrew, Lingala, Bambara, Swahili, Sepeti, Igbo, Kinyarwanda, Hausa

**Low-Resource Languages Dataset**   Given the extremely limited data for these low-resource languages, we extracted 2,000 samples from English data and used Google Translate to create training sets for each language.

**Low-Resource Languages Benchmark**   For low-resource languages with limited evaluations, following multilingual evaluation method of Llama3 (Dubey et al., 2024) we translate medical-clinical part of MMLU (Hendrycks et al., 2020) questions and answers using Google Translate. We employed a random selection of 3-shot queries to pose questions to the model, followed by answer extraction and evaluation of the model's responses.

A.6    VERIFICATION OF THE EFFECTIVENESS OF GOOGLE TRANSLATE IN MULTILINGUAL MEDICAL TRANSLATION

Table 9: Comparison of GPT-4o and Apollo-MoE Performance on MMLU Medical Sections

| Model | Ar | De | En | Fr | Hi | It | Pt | Zh | Avg. |
|---|---|---|---|---|---|---|---|---|---|
| **GPT-4o (Medical part of MMLU)** | 81.4 | 91.5 | 91.3 | 88.3 | 87.5 | 87.2 | 90.2 | 87.2 | 88.2 |
| **GPT-4o (Translated Medical part of MMLU)** | 80.9 | 90.5 | 91.3 | 89.6 | 86.9 | 87.4 | 90.8 | 87.2 | 88.2 |
| **Difference** | 0.5 | 1.0 | 0.0 | -1.3 | 0.6 | -0.2 | -0.6 | 0.0 | 0.0 |
| **Apollo-MoE (Medical part of MMLU)** | 57.9 | 73.8 | 74.8 | 71.5 | 57.2 | 72.2 | 73.4 | 84.2 | 68.7 |
| **Apollo-MoE (Translated Medical part of MMLU)** | 58.3 | 73.5 | 74.8 | 72.4 | 56.9 | 71.9 | 73.8 | 84.5 | 68.8 |
| **Difference** | -0.4 | 0.3 | 0.0 | -0.9 | 0.3 | 0.3 | -0.4 | -0.3 | -0.1 |

To verify the accuracy of the translated datasets as suggested, we sampled 100 items from the evaluation sets of two rare languages, Croatian (Hr) and Danish (Da). With the assistance of two Chinese translation

professionals in the respective languages, we enlisted three licensed Chinese doctors to annotate the accuracy of the items. The average accuracy rates were 90% and 92%, respectively, partially validating the quality of the translated datasets. Additionally, a prior study (Taira et al., 2021) demonstrated that Google Translate achieves an average accuracy of 82.5% in the complex and rigorous context of emergency medicine.

Meanwhile, for the languages evaluated using MMLU medical parts for evaluation and within MMMLU, we added the accuracy of GPT-4o and Apollo-MoE (develop from Qwen2-7B) on the MMMLU medical part (proofread by OpenAI) and the MMLU translated medical part, and compared the results. As shown in the Tab.9, the models achieved nearly identical scores on these two datasets, with the largest accuracy difference being only 1.3%.

## B    GENERAL EXPERIMENT SETUP

In this section, we describe the precise setup for our dense and MoE models. In this work, all our experiments follow the settings described below, with any specific settings mentioned directly in the main text.

**Training** All model use AdamW, a batch size of 32, a sequence length of 4096 and a cosine decay learning rate schedule with a linear warmup of proportion 0.3 for training on 8x NVIDIA A800-SXM4-80GB GPUs. For model initialization and data sampling, we set the random seed to 42. For MoE models, We fine-tune them with the same settings as before after sparse upcycling from a dense model. For all routing strategy in MoE models, router parameters are initialized randomly with a zero-mean normal distribution with standard deviation 0.02. We set the learning rate of the dense model to 1e-4 and the learning rate of the MoE model to 1e-5.

**Evaluation** To measure model performance, we extract the options from model output and calculate the accuracy with the reference answer. All evaluations use 3-shot examples. The optimal value is selected based on the average accuracy across three tests for each benchmark.

## C    PROMPTS

### C.1    PROMPT FOR GENERATING DOCTOR-PATIENT DIALOGUES

Prompts for generating doctor-patient dialogues is shown in Fig.7.

---

**Prompt**

<text>{text}</text>
Please create some dialogues between patients and doctors in English based on the above text. The format is:
<Patient>Patient's question</Patient>
<Doctor>Doctor's answer</Doctor>
Both patient questions and doctor responses are as complex and detailed as possible.

---

Figure 7: Prompt Template for Generating Doctor-Patient Dialogues

### C.2    PROMPTS FOR GENERATING QA PAIRS FROM TEXTS

Prompts for generating QA pairs from texts is shown in Fig.8.

> **Prompts**
>
> **Prompt for Generating Question:**
> Please create a <question >that closely aligns with the provided <text>. Ensure that the <question>is formulated in English and does not explicitly reference the text. You may incorporate specific scenarios or contexts in the <question>, allowing the <text>to serve as a comprehensive and precise answer.
> <text>: {text}
> <question>:
>
> **Prompt for Generating Answer:**
> You are Apollo, equipped with in-depth knowledge in medicine. Your task is to directly answer the user's <question>in English. In formulating your response, you must thoughtfully reference the <reference text>, ensuring that your reply does not disclose your reliance on <reference text>. Aim to provide a comprehensive and informative response, incorporating relevant insights from <reference text>to best assist the user. Please be cautious to avoid including any content that might raise ethical concerns.
> <question>: {question}
> <reference text>: {reference}
> <reply>:

Figure 8: Prompts for Generating QA Pairs from Texts. We show the English version of Prompt, and other languages are similar.

## D    TOKEN EXPERT ROUTING CONSTRUCTION

Due to the varying number of tokens required for different languages, sentences of the same length may have different token counts depending on the language. We randomly selected varying amounts of data from benchmarks in different languages. After removing common English characters, the data was used as a test set for expert routing analysis, ensuring that each language had 80,000 tokens. Fig. 9 shows sample test data for several languages; the format is the same for other languages.

## E    ADVANTAGES OF GENERALIZABILITY

The primary method proposed in this paper, PostMoE with language family experts, offers the advantages of generalizability, enabling adding other languages efficiently and effectively.

**Experiments** To provide a more intuitive understanding, we selected two additional languages not included in the 50 languages: *Bengali* (Bn) and *Amharic* (Am), which belong to the Indo-European and Afro-Asiatic language families respectively, to demonstrate the model's efficient generalization capability. Specifically, we processed 2,000 data samples for *Bengali* and *Amharic* and continued fine-tuning the Dense models and ApolloMoE models.

**Results** As shown in Tab.10, the experimental results demonstrate the clear advantages of the proposed method in adapting to additional languages. For performance improvement in newly added languages, the ApolloMoE model outperforms the Dense model across all scales, with training-related gains also showing advantages in most cases. For preserving the performance of original languages, the ApolloMoE model maintains or even improves performance across nearly all scales, whereas the Dense model generally experiences a decline in performance.

---

**Examples**

**English**
You are a medical doctor answering real-world medical exam questions. Select one correct answer. Question: Diagnosis of iron deficiency can be complicated by concurrent infection since many markers of iron status are altered by infection. Which of the following combinations of iron status markers is likely to be found in a person with both iron deficiency and a severe infection?
Options:
Low haemoglobin, high ferritin, high serum transferrin receptors, high hepcidin
Low haemoglobin, low ferritin, high serum transferrin receptors, low hepcidin
Low haemoglobin, low ferritin, normal serum transferrin receptors, high hepcidin
Low haemoglobin, low ferritin, low serum transferrin receptors, high hepcidin
The correct answer is Low haemoglobin, high ferritin, high serum transferrin receptors, high hepcidin.

**French**
Vous êtes un médecin et répondez à des questions d'examen médical du monde réel. Veuillez choisir une bonne réponse.
question: Quelle est la pathologie qui s'accompagne d'un hypercorticisme?
Possibilités:
Maladie d'Addison.
Maladie de Cushing.
Syndrome de Conn.
Maladie de Basedow.
Syndrome de Barterr.
La bonne réponse est Maladie de Cushing.

**Portuguese**
Você é um médico que responde a perguntas de exames médicos do mundo real. Escolha uma resposta correta
pergunta:carga de energia da célula é:
Opções:
a diferença entre a carga no exterior e no interior de uma célula.
gerado pela ATPase sódio-potássio.
a taxa geral de uso de energia pela célula
o grau em que o pool total de nucleotídeos de adenina está fosforilado.
A resposta correta é o grau em que o pool total de nucleotídeos de adenina está

**Germany**
Sie sind ein Arzt, der Fragen zu medizinischen Untersuchungen aus der Praxis beantwortet. Bitte wählen Sie eine richtige Antwort
Frage: Was erklärt am besten, wie Mutationen in der DNA zu einer neuen Phänotyp-Expression führen können?
Optionen:
Ein anderes Polypeptid wird produziert.
Die Polarität von tRNA wird das Gegenteil von der von DNA.
Nukleinsäuren sind methyliert.
Das Gen wird jetzt in Richtung 3' bis 5' gelesen.
Die richtige Antwort ist Ein anderes Polypeptid wird produziert.

Figure 9: Test Data Example in Different Languages. Other Languages are Similar.

Table 10: Continue Fine-tuning with Bn and Am Data Based on Dense and PostMoE Models.

| Models | Avg. Accuracy | | Lang-Spec. Accuracy | |
|---|---|---|---|---|
| | **High** | **Low** | **Bn** | **Am** |
| **Based on Qwen2-0.5B** | | | | |
| **Dense** | 39.2 | 33.2 | 36.4 | 31.5 |
| +Bn | 38.2 (-1.0) | 33.1 (-0.1) | 37.5 (+0.9) | - |
| +Am | 38.1 (-1.1) | 32.9 (-0.3) | - | 33.9 (+2.4) |
| **ApolloMoE** | 40.5 | 34.6 | 37.8 | 33.0 |
| +Bn | 40.6 **(+0.1)** | 34.6 **(-0.0)** | **39.7 (+1.9)** | - |
| +Am | 40.5 **(-0.0)** | 34.6 **(-0.0)** | - | **36.0 (+3.0)** |
| **Based on Qwen2-1.5B** | | | | |
| **Dense** | 52.2 | 43.7 | 44.0 | 35.0 |
| +Bn | 49.1 (-3.1) | 39.2 (-4.5) | 50.8 **(+6.8)** | - |
| +Am | 47.7 (-4.5) | 39.1 (-4.6) | - | 36.4 (+1.4) |
| **ApolloMoE** | 54.8 | 44.9 | 50.4 | **38.6** |
| +Bn | 54.0 **(-0.8)** | 44.9 **(-0.0)** | **55.7** (+5.3) | - |
| +Am | 54.8 **(-0.0)** | 44.9 **(-0.0)** | - | **41.9 (+3.3)** |
| **Based on Qwen2-7B** | | | | |
| **Dense** | 69.0 | 56.7 | 66.3 | 35.7 |
| +Bn | 68.4 (-1.0) | 55.7 (-1.0) | 68.9 **(+2.6)** | - |
| +Am | 68.3 (-1.1) | 55.3 (-1.4) | - | 38.2 (+2.5) |
| **ApolloMoE** | 69.9 | 58.3 | 67.1 | 40.5 |
| +Bn | 69.6 **(-0.3)** | 58.5 **(+0.2)** | **69.5**(+2.4) | - |
| +Am | 69.5 **(-0.4)** | 58.3 **(-0.0)** | - | **42.5 (+2.5)** |

## F    MECHANISTIC INTERPRETABILITY

**Mechanistic Interpretability** Circuit analysis has emerged as a new paradigm of model analysis, providing an in-depth mechanistic interpretability of the internal information flow and hierarchical structure of models (Olah et al., 2020b; Merullo et al., 2024). Additionally, the study of modularity (Meng et al., 2022; Wang et al., 2022) and sparcity (Olsson et al., 2022) within models offers insights into constructing specialized submodules, which helps to organize information sharing and feature isolation more effectively. Component-level detailed analysis (Olsson et al., 2022; Vig et al., 2020; Singh et al., 2024) delves into the contributions of attention heads, neurons, and other components to model behavior. Lastly, causal mechanism analysis (Vig et al., 2020; Geiger et al., 2024) provides a method for explaining the relationships between information flow and functionality, helping to uncover the key mechanisms within models.

## G    DIFFERENT LANGUAGES FOR HYBRID ROUTING

The Hybrid routing distribution is shown in Fig. 10. It explains how the model allocates tasks of processing different languages among its internal experts, and how this allocation changes across different network layers.

## H    COMPLETE RESULTS ON 12 HIGH-RESOURCE AND 38 LOW-RESOURCE LANGUAGES

To comprehensively evaluate the performance of the proposed model across different languages, Tab. 11 presents the accuracy results on 12 high-resource languages and 38 low-resource languages. We showcase the performance of MoE models with various routings (in Sec.3.1.2) and the Apollo-MoE series across different base model sizes (e.g. 0.5B, 1.5B, and 7B parameters in Sec.4) on these languages. The models achieve accuracy on par with that of the high-resource languages, even for low-resource languages, highlighting the model's strong generalization capabilities when handling resource-scarce languages.

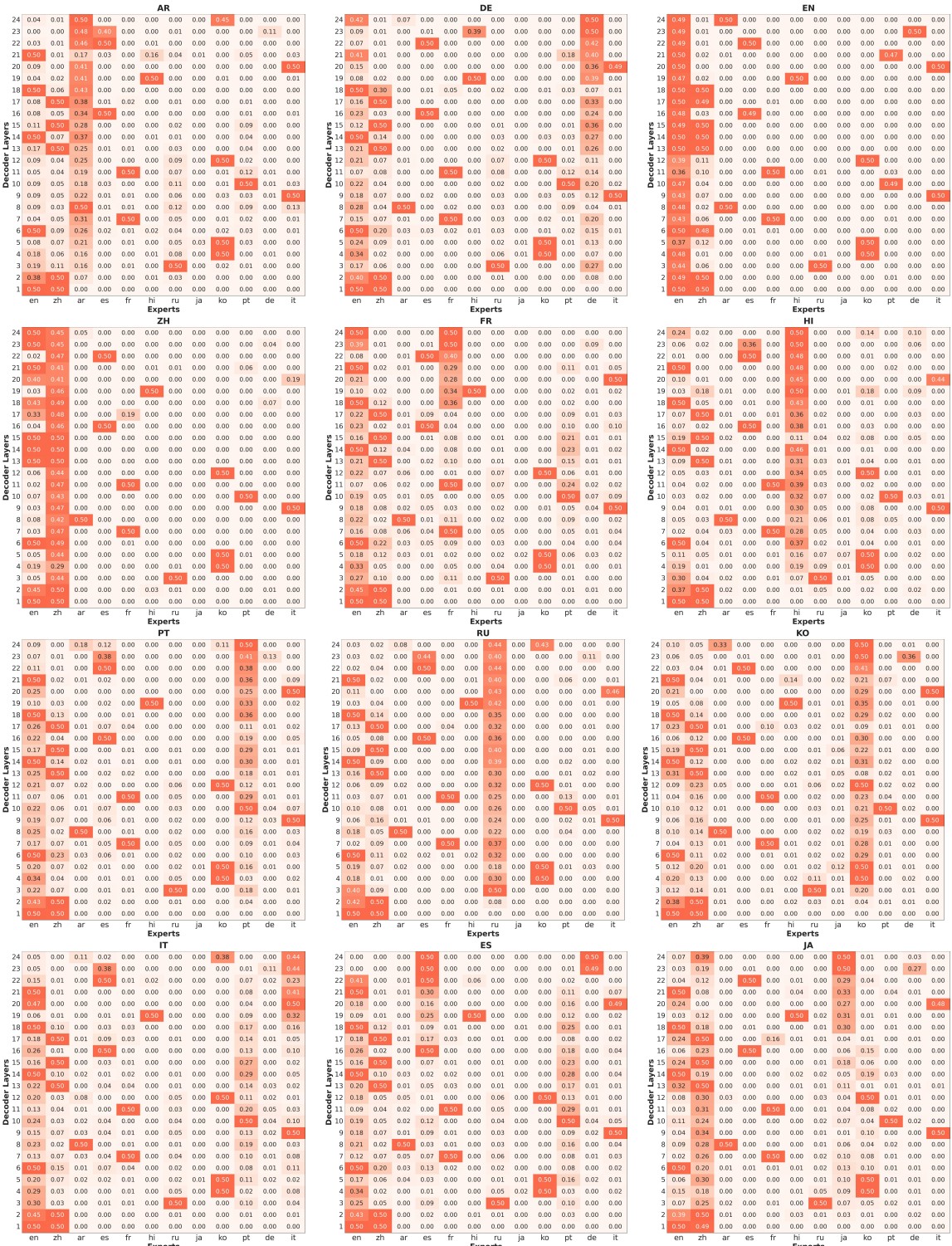

Figure 10: Different Languages for Hybrid Routing.

Table 11: Vital Model Performances (Accuracy) on High- and Low-Resource Languages.

| Langs | Qwen2-0.5B | Lang-Spec. | Top2 | Hybrid2 | Apollo-MoE-0.5B | Apollo-MoE-1.5B | Apollo-MoE-7B |
|---|---|---|---|---|---|---|---|
| Zh | 45.4 | 42.8 | 53.2 | 53.6 | 51.3 | 67.8 | 84.1 |
| Ko | 19.2 | 27.8 | 35.9 | 36.2 | 35.2 | 52.9 | 68.4 |
| Ja | 24.0 | 21.4 | 30.5 | 32.4 | 26.9 | 45.7 | 62.4 |
| Ne | 29.2 | 23.9 | 29.2 | 29.2 | 27.3 | 40.2 | 56.1 |
| Th | 33.4 | 32.2 | 26.1 | 28.8 | 39.0 | 45.5 | 59.8 |
| Vi | 34.1 | 27.1 | 30.9 | 29.1 | 35.9 | 45.6 | 61.1 |
| Lo | 34.1 | 27.3 | 28.0 | 30.3 | 34.1 | 37.5 | 45.8 |
| Mg | 30.7 | 22.7 | 27.7 | 34.1 | 32.2 | 41.7 | 46.2 |
| Ceb | 36.7 | 26.5 | 28.8 | 35.2 | 32.2 | 50.8 | 67.0 |
| Su | 32.2 | 30.7 | 33.0 | 35.2 | 34.8 | 48.5 | 67.8 |
| Ilo | 29.2 | 26.1 | 26.1 | 28.8 | 32.6 | 46.2 | 61.0 |
| Doi | 27.3 | 24.6 | 22.7 | 23.5 | 29.9 | 41.3 | 50.8 |
| En | 39.3 | 39.1 | 43.7 | 44.1 | 45.4 | 56.5 | 73.1 |
| De | 27.4 | 28.6 | 36.9 | 37.2 | 38.2 | 54.8 | 73.5 |
| Pt | 26.3 | 31.0 | 34.8 | 34.3 | 37.3 | 57.3 | 73.8 |
| Es | 32.9 | 33.3 | 38.9 | 40.0 | 39.8 | 53.5 | 69.4 |
| Fr | 21.3 | 16.5 | 39.9 | 40.7 | 38.4 | 53.3 | 72.4 |
| Ru | 46.9 | 52.3 | 58.2 | 58.8 | 64.1 | 72.9 | 74.2 |
| It | 20.7 | 24.5 | 37.7 | 38.9 | 39.9 | 54.4 | 71.9 |
| Hr | 32.6 | 26.9 | 31.8 | 34.1 | 35.6 | 48.9 | 67.0 |
| Gl | 36.0 | 23.9 | 37.9 | 38.6 | 40.2 | 56.8 | 71.6 |
| Cs | 34.5 | 26.9 | 32.6 | 32.6 | 36.4 | 54.2 | 67.0 |
| Co | 35.2 | 20.8 | 36.0 | 35.2 | 44.3 | 56.1 | 70.8 |
| La | 30.3 | 20.8 | 29.5 | 34.8 | 37.5 | 48.5 | 60.6 |
| Uk | 31.5 | 26.5 | 33.5 | 30.9 | 29.0 | 35.4 | 53.3 |
| Bs | 31.1 | 25.0 | 32.2 | 36.4 | 37.9 | 48.9 | 64.0 |
| Bg | 31.1 | 28.8 | 35.6 | 36.0 | 31.1 | 41.7 | 60.2 |
| Eo | 28.8 | 22.0 | 28.4 | 31.4 | 34.5 | 45.8 | 62.9 |
| Mai | 28.8 | 25.0 | 26.1 | 28.0 | 28.8 | 42.8 | 59.8 |
| Sq | 28.4 | 22.0 | 30.3 | 30.7 | 31.4 | 40.5 | 58.3 |
| Da | 33.3 | 25.4 | 30.7 | 36.4 | 36.7 | 55.3 | 67.8 |
| Sa | 29.2 | 28.0 | 27.3 | 30.3 | 25.0 | 40.2 | 54.5 |
| No | 33.7 | 25.8 | 27.3 | 31.8 | 33.7 | 47.3 | 67.8 |
| Gn | 30.7 | 25.4 | 28.0 | 36.0 | 34.1 | 45.8 | 56.1 |
| Sr | 30.7 | 28.0 | 33.0 | 34.8 | 17.8 | 33.7 | 39.4 |
| Sk | 34.1 | 27.7 | 35.2 | 34.1 | 38.3 | 50.8 | 67.0 |
| Gd | 31.8 | 23.1 | 29.9 | 30.3 | 33.7 | 45.8 | 50.8 |
| Lb | 35.2 | 24.6 | 28.8 | 30.7 | 38.3 | 54.2 | 67.8 |
| Hi | 32.9 | 33.3 | 38.9 | 40.2 | 39.8 | 53.5 | 69.4 |
| Ar | 27.3 | 28.8 | 34.5 | 35.1 | 36.3 | 47.2 | 58.3 |
| Ckb | 26.9 | 26.1 | 31.8 | 31.8 | 31.8 | 37.9 | 48.1 |
| Mt | 25.8 | 24.6 | 32.2 | 30.3 | 25.0 | 47.7 | 60.2 |
| He | 34.1 | 27.9 | 29.0 | 32.2 | 37.0 | 43.5 | 64.0 |
| Ln | 31.8 | 26.1 | 29.9 | 27.7 | 33.7 | 43.9 | 53.4 |
| Bm | 31.1 | 28.0 | 24.2 | 29.9 | 29.2 | 42.4 | 40.2 |
| Sw | 30.7 | 27.3 | 30.3 | 33.0 | 37.5 | 42.0 | 46.2 |
| Nso | 29.5 | 29.9 | 27.3 | 31.8 | 34.5 | 40.9 | 45.8 |
| Ig | 30.3 | 27.3 | 28.0 | 30.3 | 36.0 | 44.3 | 49.6 |
| Rw | 27.3 | 27.3 | 29.9 | 29.5 | 36.7 | 34.5 | 34.1 |
| Ha | 31.1 | 31.8 | 26.5 | 30.7 | 26.9 | 39.0 | 37.1 |

