# OpenReview forum: "Efficiently Democratizing Medical LLMs for 50 Languages via a Mixture of Language Family Experts"
_ICLR.cc/2025/Conference — ICLR 2025 Poster_

### Official Review · Reviewer_CPZy · 2024-10-27

**Soundness:** 4
**Presentation:** 4
**Contribution:** 3
**Rating:** 8
**Confidence:** 4

**Summary:**

In this paper, the authors detail with great precision the training procedure of their Mixture-of-Language-Family-Experts LLM for the medical domain, which combines several existing techniques in a novel way. The authors balance the benefits of cross-lingual learning (making sure low-resource languages benefit from knowledge only written in high-resource languages) with the curse of multilinguality (cross-lingual contamination of linguistic structures) using linguistic experts which always activate for their respective language but can also activate for other languages at the router's discretion, thereby ensuring that sentences in the train set benefit multiple languages simultaneously without impacting all aspects of every other languages. An evaluation is carried for 12+ languages (up to 50 in the scale-up experiments).

**Strengths:**

- Extensive and multi-layered analysis of cross-lingual pollination and the various trade-offs involved both during training and at inference time.
- High-quality design choices informed by serious and well-informed model analyses.
- Well-structured and information-rich paper.
- On top of that: an impressive new model for the medical domain.

**Weaknesses:**

- [addressed] Evaluation is performed on translated test sets, generated by other language models.
- [addressed] Little discussion is made of the risk of test set contamination in the training set of the model in languages other than the source language. The 64 token overlap used breaks down once the input data is multilingual, since translations have no or minimal token overlap.
- Most techniques presented in the paper do not appear to be very novel, even though their combination seems to be, and their choices are well-justified. Still, it's sometimes tough for me to determine whether this is really an academic work or just a high-quality engineering report written in paper form.

**Questions:**

- [addressed] Considering that the language specialization of LLMs at the top and bottom layers was an arealdy-known phenomenon by early 2024 (e.g. "Do Llamas Work in English? On the Latent Language of Multilingual Transformers" or "How do Large Language Models Handle Multilingualism?"), do you still perceive that your "Spread Out in the End" analysis is bringing novel insights? In general, could you help me perceive the theorical aspects of your other contributions which I may have missed?
- [addressed] Did you release your multilingual test set to enable other works to evaluate against yours?
- [addressed] Do you intend to release your training data to facilitate replications of this work?
- What role does relative importance of languages play in the linguistic family grouping? English is likely overshadowing other languages using the same expert.
- [addressed] How did you choose your 50 languages? Moreover, why did you include Latin in your experiments? It's no longer an active language of science since many centuries, and Google Translate is unlikely to do a good job on modern medical documents in this case.

Not technical but still relevant:
- [addressed] Why do you want to present this work at ICLR? As a first-time ICLR review, I cannot make judgements about what type of works usually make an impact at ICLR, so I will leave this up to the AC Meta-Review. I will however raise my concern that this work doesn't seem to have a particularly strong tie to Representation Learning. It's also not particularly analytical or theoretical. While I personally would be very excited to discuss this further with the authors, I'm surprised to see this type of work submitted to ICLR instead of EMNLP or COLM. However, I see that ICLR also includes general machine learning as an accepted topic, so it's really difficult for me to judge this, and I won't. But I think the answer to this question might be helpful for the meta-reviewer.

---

> ### Author Response · Authors · 2024-11-25
> **Thanks for your kind reviews (1/N)**
>
> Thank you for your kind and insightful comments. This paper aims to democratize medical AI for a broader linguistic community. By collecting data, analyzing and leveraging the model's multilingual mechanisms, and integrating linguistic priors, it provides the community with high-quality data, efficient, generalizable, and scalable technical solutions, as well as high-performing models. Below, we will address your questions one by one.
>
> > **Q1: Evaluation is performed on translated test sets, generated by other language models.**
>
> Regarding the evaluation, we used **peer-reviewed medical evaluation datasets**, which are detailed in **Appendix A.3** and **A.5** and referenced in lines 126-127 and 226-227 of the revised version. For languages without existing medical evaluation datasets, we utilized the **medical-related sections of MMLU** to create evaluation sets for those languages. In total, we integrated **10** medical evaluation datasets for the corresponding languages and supplemented others with the medical section of MMLU for other languages. The specific datasets are described in detail in **Table 7**.
>
> To **further verify the accuracy of the translated datasets as suggested**, we sampled 100 items from the evaluation sets of two minor languages, Croatian (Hr) and Danish (Da). With the assistance of two Chinese translation professionals in the respective languages, we enlisted three licensed Chinese doctors to annotate the accuracy of the items. The average accuracy rates were **90%** and **92%**, respectively, partially validating the quality of the translated datasets. Additionally, a prior study [1] demonstrated that Google Translate achieves an average accuracy of **82.5%** in the complex and rigorous context of emergency medicine.
>
> Meanwhile, for the languages evaluated using MMLU medical parts for evaluation and within MMMLU, we use the accuracy comparation of GPT-4o and Apollo-MoE (develop from Qwen2-7B) on the **MMMLU medical part (proofread by OpenAI) and the MMLU translated medical part**, and compared the results. As shown in the Table below, the models achieved nearly identical scores on these two datasets, with the largest accuracy difference being only 1.3\%. Further verified the effectiveness of Google Translate.
>
> The results are mentioned in the lines 127-128 and datailed in the Appendix A.6 of the revised version.
>
> |                                              | ar   | de   | en   | fr   | hi   | it   | pt   | zh   | Avg  |
> | -------------------------------------------- | ---- | ---- | ---- | ---- | ---- | ---- | ---- | ---- | ---- |
> | Gpt4-o (Medical part of mmmlu)               | 81.4 | 91.5 | 91.3 | 88.3 | 87.5 | 87.2 | 90.2 | 87.2 | 88.2 |
> | Gpt4-o (Translated Medical part of mmlu)     | 80.9 | 90.5 | 91.3 | 89.6 | 86.9 | 87.4 | 90.8 | 87.2 | 88.2 |
> | Difference                                   | 0.5  | 1.0  | 0.0  | -1.3 | 0.6  | -0.2 | -0.6 | 0.0  | 0.0  |
> | &nbsp;                                       |      |      |      |      |      |      |      |      |      |
> | Apollo-MoE (Medical part of mmmlu)           | 57.9 | 73.8 | 74.8 | 71.5 | 57.2 | 72.2 | 73.4 | 84.2 | 68.7 |
> | Apollo-MoE (Translated Medical part of mmlu) | 58.3 | 73.5 | 74.8 | 72.4 | 56.9 | 71.9 | 73.8 | 84.5 | 68.8 |
> | Difference                                   | -0.4 | 0.3  | 0.0  | -0.9 | 0.3  | 0.3  | -0.4 | -0.3 | -0.1 |
>
> However, we also acknowledge that translation is a stopgap measure to showcase the medical multilingual capabilities of Models the community. In future work, we plan to leverage the influence of this study to seek support from local medical professionals and focus on building a multilingual medical evaluation set with localized characteristics. This will be a tedious but meaningful work.
>
>
>
> > **Q2: Little discussion is made of the risk of test set contamination in the training set of the model in languages other than the source language.**
>
> For other minor languages, we extracted 2,000 samples from the English data and used Google Translate to create training corpora for each language. Since the source of the translation data inherently eliminates the risk of test set leakage, the risk of leakage in the translated dataset is minimal.
>
> Moreover, to address your concerns, we conducted data leakage detection on Croatian (Hr) and Danish (Da) using a 16-token overlap as the measurement criterion as suggested. This process identified 2 (**0.1%**) and 0 (**0.0%**) overlapping data items out of 2000, respectively, further supporting the above statement.
>
> [1] [A Pragmatic Assessment of Google Translate for Emergency Department Instructions](https://pubmed.ncbi.nlm.nih.gov/33674922/)

---

> ### Author Response · Authors · 2024-11-25
> **Thanks for your kind reviews (2/N)**
>
> > **Q3: Most techniques presented in the paper do not appear to be very novel, even though their combination seems to be, and their choices are well-justified.**
>
> Complexity is not the goal. Towards addressing real practical problems, simpler methods are often more elegant and can provide valuable insights to the community. We use Hybrid Routing as a tool to explore model mechanisms. By employing Post-MoE and introducing language-family experts, we effectively utilize the discovered mechanisms to efficiently democratize medical AI for a broader linguistic community.
>
>
>
>
>
> > **Q4: In general, could you help me perceive the theorical aspects of your other contributions which I may have missed?**
>
> As noted in the paper, we have observed and drawn inspiration from studies on neural analysis that explore large-model multilingual mechanisms. From **an application perspective**, our method leverages the modular characteristics of MoE to efficiently utilize the discovered mechanisms. It can be extended to accommodate any number of languages while enabling the combination and disassembly of modules.
>
> **Theoretically**, while many studies have highlighted the presence of multilingual neurons at the bottom layers of the model, we further observed that activation parameters at bottom layers are concentrated in the Chinese and English language modules rather than being distributed across various language modules. We hypothesize that this may reflect a model strategy that relies on core language experts at the bottom layers for multilingual translation. As this finding is not directly relevant to the main focus of the paper and is a broader multilingual question, we chose not to include it explicitly and plan to explore it in future work.
>
> This observation also demonstrates that **our method provides an additional dimension (language experts)** for analyzing multilingual mechanisms beyond the depth perspective. It invites further investigation into questions such as which language experts should be utilized at specific depths and how language experts at different depths interact with one another.
>
> **Moreover**, by treating different modalities as distinct languages, the code framework provided in this paper can also be applied to study how modalities interact during the multimodal adaptation process.
>
>
>
> > **Q5: What role does relative importance of languages play in the linguistic family grouping? English is likely overshadowing other languages using the same expert.**
>
> As shown in Table 1, **different languages generally enhance each other**, with English having **a positive impact** on other languages. Your perspective is certainly valuable, and we plan to conduct more fine-grained analyses in future works.
>
>
>
> > **Q6:  How did you choose your 50 languages? Moreover, why did you include Latin in your experiments? It's no longer an active language of science since many centuries, and Google Translate is unlikely to do a good job on modern medical documents in this case.**
>
> We selected representative languages **based on geographical distribution**. Latin was included because much **traditional Western medical knowledge is recorded in Latin.** Additionally, Latin is widely used in biology, anatomy, and medical law. Teaching the model to understand Latin can assist the community in interpreting and drawing inspiration from the knowledge of predecessors.
>
>
>
> > **Q7: About Open source**
>
> Open sourcing is actually a core motivation of our work. We have released all datasets, evaluation sets, code, and models from the paper, which have received significant downloads.
>
>
>
> > **Q8: Why do you want to present this work at ICLR?**
>
> Frankly speaking, ICLR has a broader influence. Given the need for the work to reach a larger audience and advance related research fields by engaging more like-minded researchers, we chose this conference.
>
> Additionally, ICLR has featured significant related work in recent years on topics such as multilingual models [1], MoE architectures [2][3], and medical applications [4].
>
> &emsp;
>
> [1] [Large Multilingual Models Pivot Zero-Shot Multimodal Learning across Languages](https://arxiv.org/abs/2308.12038)
>
> [2] [Sparse Upcycling: Training Mixture-of-Experts from Dense Checkpoints](https://arxiv.org/abs/2212.05055)
>
> [3] [Sparse MoE with Language Guided Routing for Multilingual Machine Translation](https://openreview.net/forum?id=ySS7hH1smL)
>
> [4] [Medical Image Understanding with Pretrained Vision Language Models: A Comprehensive Study](https://arxiv.org/abs/2209.15517)

---

> > ### Comment · Reviewer_CPZy · 2024-11-27
> >
> > Thank you for your revision of the paper and detailed response. Given that you addressed very exhaustively my concerns about test set translation and cross-lingual data contamination, and in light of your explanation of the theoretical contributions of your paper, I have decided to increase my contribution score to 3 (good), as well as my overall score to a clearer acceptance score. As a score of 7 is not possible to give, I made the decision to update my score to 8, because I believe this paper ought to have an average score above 6. I would still encourage the authors to make minor updates in their camera ready to better highlight their theoretical contributions.

---

> > > ### Author Response · Authors · 2024-11-28
> > >
> > > Thank you for your support, it means a lot to us. we will highlight our theoretical contributions as suggested.

---

### Official Review · Reviewer_sQUP · 2024-11-02

**Soundness:** 3
**Presentation:** 3
**Contribution:** 3
**Rating:** 6
**Confidence:** 3

**Summary:**

The paper introduces a method to expand medical language models to cover 50 languages, focusing on those with limited data. Instead of scaling by adding experts for each language, the authors group languages into families, using shared experts to keep the model size manageable. They identify a key insight: early model layers handle cross-language information, while later layers focus on individual languages.

Based on this, they propose a design that only applies sparse routing in the final layers, enhancing efficiency. Experiments show the model works well across both major and minor languages, suggesting it could improve access to medical information in diverse languages without requiring excessive data or computational costs.

**Strengths:**

Comprehensive experiments are conducted across models with varying parameter sizes (0.5B, 1.5B, and 7B), demonstrating scalability and superior performance among major languages.

The “Spread Out in the End” phenomenon is well-motivated and appears relevant for improving both interpretability and efficiency in multilingual models.

**Weaknesses:**

Although the MoE model's efficiency is emphasized, a more extensive comparison with the same computation cost dense models (especially regarding computational cost and training efficiency) could strengthen the claims.

The improvement mostly appears in Italian and French, also decreased in many more low-resource languages. This seems to lead to some overclaim... how is the data-mixture determined?

line 124: employing Google Translate to translate the questions and answers of MMLU - why not use mmmlu from OpenAI?

Have any experiments been conducted to assess the robustness of translations used for low-resource languages? If not, how do you propose to validate the accuracy of these translations, given the potential consequences in a medical context?

**Questions:**

Fig 3b is really hard to read

 #params is more appropriate and apple-to-apple comparison with same active parameters comparing to dense models?

---

> ### Author Response · Authors · 2024-11-25
> **Thanks for your kind reviews (1/N)**
>
> Thank you for your kind and insightful reviews. Below, we will address your questions one by one.
>
> > **Q1: Although the MoE model's efficiency is emphasized, a more extensive comparison with the same computation cost dense models (especially regarding computational cost and training efficiency) could strengthen the claims.**
>
> Thank you for your suggestion, which will indeed enhance the robustness of the paper. We have conducted additional experiments based on your recommendations and will present them below.
>
> **Experiment Settings**: To accurately construct a dense model of equivalent size, we replicated the MLP following the approach used in MoE Upcycling and initialized the routing with an average distribution. Unlike MoE, the initialized dense model employs full activation instead of sparse activation.
>
> - **Table 2 Related Supplementary Experiments**:
>
>   The purpose of Table 2 is to demonstrate the generalization advantage of the Hybrid Routing method over other approaches. We have provided **supplementary baseline results** for dense models with parameters matching either the total parameters and the activation parameters of the MoE models. As shown in the table, compared to a Dense model with the same total parameters, the MoE model exhibits better generalization on minor languages. Additionally, compared to a Dense model with the same activation parameters, the MoE model performs better on both major and minor languages. For related content, please refer to lines 195 and 222 in new version paper.
>
>   | Method/Model                   | Active Param. | Param.    | Avg. Major | Avg. Minor |
>   | ------------------------------ | ------------- | --------- | ---------- | ---------- |
>   | Qwen2-0.5B                     | 0.49B         | 0.49B     | 29.7       | 31.5       |
>   | Dense                          | 0.49B         | 0.49B     | 37.8       | 24.6       |
>   | Dense with Same  Active Param. | **0.81B**     | 0.81B     | 38.4       | 26.2       |
>   | Dense with  Same Total Param.       | 3.95B         | **3.95B** | **42.0**   | 30.9       |
>   | MoE with Hybrid-k Routing                       | **0.81B**     | **3.95B** | 40.0       | **32.0**   |
>
> - **Table 5 Related Supplementary Experiments**:
>
>   The purpose of Table 5 is to evaluate the scalability of the primary method proposed in this paper (PostMoE with language family experts). Due to time constraints, we only supplemented the dense baselines with the same activation parameters. As shown in the Table, the MoE models outperform the Dense model with the same number of activation parameters. For related content, please refer to line 394 in new version paper.
>
>   | Method/BaseModel               | Active Param. | Avg. Major | Avg. Minor |
>   | ------------------------------ | ------------- | ---------- | ---------- |
>   | **Qwen2-0.5B**                 | 0.49B         | 29.7       | 31.5       |
>   | Dense                          | 0.49B         | 39.2       | 32.2       |
>   | Dense with Same  Active Param. | 0.52B         | 39.4       | 34.0       |
>   | MoE with Hybrid-k Routing      | 0.52B         | **40.5**   | **34.6**   |
>   | &nbsp;                         |               |            |            |
>   | **Qwen2-1.5B**                 | 1.54B         | 42.9       | 38.4       |
>   | Dense                          | 1.54B         | 52.2       | 43.7       |
>   | Dense with Same  Active Param. | 1.63B         | 52.8       | 44.1       |
>   | MoE with Hybrid-k Routing      | 1.63B         | **54.8**   | **44.9**   |
>   | &nbsp;                         |               |            |            |
>   | **Qwen2-7B**                   | 7.62B         | 55.2       | 49.2       |
>   | Dense                          | 7.62B         | 69.0       | 55.7       |
>   | Dense with Same  Active Param. | 8.02B         | 68.5       | 56.3       |
>   | MoE with Hybrid-k Routing      | 8.02B         | **69.9**   | **58.3**   |
>
> > **Q2: The improvement mostly appears in Italian and French, also decreased in many more low-resource languages. This seems to lead to some overclaim... how is the data-mixture determined?**
>
> In Tables 2 and 4, the model performance improved across all languages after training, with an average increase **exceeding 10%**. In Table 5, all minor languages showed improvement except for Serbian (Sr), and **nearly one-third** achieved an average improvement of **over 10%**. For Serbian, we observed the performance decline also within Dense models, which may be attributed to the unique characteristics of its alphabet and linguistic logic.
>
> Regarding data mixing, we collected, processed, and utilized as much high-quality medical data as possible without deliberately explore the data-mixture influence. However, we agree that this is a highly promising research direction, and we appreciate your insightful suggestion.

---

> > ### Author Response · Authors · 2024-11-29
> > **Follow-Up on Review and Feedback**
> >
> > Dear Reviewer **sQUP**,
> >
> > We hope this message finds you well.
> >
> > We have carefully addressed all your questions and concerns, including conducting additional experiments as requested, and have provided detailed responses in the rebuttal.
> >
> > As the rebuttal deadline is approaching, we would deeply appreciate it if you could share your updated thoughts based on the rebuttal and paper revision, or do not hesitate to let us know if you have additional questions, and we will respond promptly.
> >
> > Thank you again for your thoughtful review and your invaluable contributions to the quality of this paper.
> >
> > Kind regards,
> >
> > Paper 10250 Authors

---

> ### Author Response · Authors · 2024-11-25
> **Thanks for your kind reviews (2/N)**
>
> > **Q3: employing Google Translate to translate the questions and answers of MMLU - why not use mmmlu from OpenAI?**
>
> OpenAI released MMMLU around September 23, 2024, very close to the conference deadline of October 1, 2024. Unfortunately we were not able to use it. Thank you for giving us the information.
>
>
> > **Q4: Have any experiments been conducted to assess the robustness of translations used for low-resource languages? If not, how do you propose to validate the accuracy of these translations, given the potential consequences in a medical context?**
>
> For evaluation, we utilized a peer-reviewed medical evaluation dataset, detailed in Appendix A.3 and A.5. For languages lacking existing medical evaluation datasets, we followed the LLaMA3 multilingual evaluation framework and used the medical-related sections of Google Translate MMLU to construct evaluation sets for these languages.
>
> To **verify the accuracy of the translated datasets as suggested**, we sampled 100 items from the evaluation sets of two rare languages, Croatian (Hr) and Danish (Da). With the assistance of two Chinese translation professionals in the respective languages, we enlisted three licensed Chinese doctors to annotate the accuracy of the items. The average accuracy rates were **90%** and **92%**, respectively, partially validating the quality of the translated datasets. Additionally, a prior study [1] demonstrated that Google Translate achieves an average accuracy of **82.5%** in the complex and rigorous context of emergency medicine.
>
> However, we also acknowledge that translation is a stopgap measure to showcase the medical multilingual capabilities of Models the community. In future work, we plan to leverage the influence of this study to seek support from local medical professionals and focus on building a multilingual medical evaluation set with localized characteristics. This will be a tedious but meaningful work.
>
> **Typo: " 3b is unclear"** We have enhanced the resolution of the image. The corresponding explanation of the image can be found in lines 250-252 and 284-289 of the revised version. We hope this will be helpful to you.
>
> &emsp;
>
> [1] [A Pragmatic Assessment of Google Translate for Emergency Department Instructions](https://pubmed.ncbi.nlm.nih.gov/33674922/)

---

> > ### Comment · Reviewer_sQUP · 2024-11-26
> > **Thank you for the responses**
> >
> > Not sure 82.5% out of 100 instances are high quality enough. Running mmmlu comparing to current score might be quite useful.

---

> ### Author Response · Authors · 2024-11-27
> **Thanks for your kind reviews**
>
> Thank you for your suggestion. Following your recommendation, for the languages evaluated using MMLU medical parts for evaluation and within MMMLU, we added the accuracy of GPT-4o and Apollo-MoE (develop from Qwen2-7B) on the MMMLU medical part and the MMLU translated medical part , and compared the results. As shown in the table, the models achieved nearly identical scores on these two datasets, with the largest accuracy difference being only **1.3%**. Additionally, as mentioned earlier, in our rigorous manual verification, **90%** and **92%** of the 100 instances were completely consistent. The results are mentioned in the lines 127-128 and datailed in the Appendix A.6 of the revised version.
>
> |                                              | ar   | de   | en   | fr   | hi   | it   | pt   | zh   | Avg  |
> | -------------------------------------------- | ---- | ---- | ---- | ---- | ---- | ---- | ---- | ---- | ---- |
> | Gpt4-o (Medical part of mmmlu)               | 81.4 | 91.5 | 91.3 | 88.3 | 87.5 | 87.2 | 90.2 | 87.2 | 88.2 |
> | Gpt4-o (Translated Medical part of mmlu)     | 80.9 | 90.5 | 91.3 | 89.6 | 86.9 | 87.4 | 90.8 | 87.2 | 88.2 |
> | Difference                                   | 0.5  | 1.0  | 0.0  | -1.3 | 0.6  | -0.2 | -0.6 | 0.0  | 0.0  |
> |              &nbsp;                                |      |      |      |      |      |      |      |      |      |
> | Apollo-MoE (Medical part of mmmlu)           | 57.9 | 73.8 | 74.8 | 71.5 | 57.2 | 72.2 | 73.4 | 84.2 | 68.7 |
> | Apollo-MoE (Translated Medical part of mmlu) | 58.3 | 73.5 | 74.8 | 72.4 | 56.9 | 71.9 | 73.8 | 84.5 | 68.8 |
> | Difference                                   | -0.4 | 0.3  | 0.0    | -0.9 | 0.3  | 0.3  | -0.4 | -0.3 | -0.1 |

---

> > ### Comment · Reviewer_sQUP · 2024-11-29
> >
> > Thanks, I have no further questions.

---

> > > ### Author Response · Authors · 2024-11-30
> > >
> > > Thanks for your kind reviews.

---

### Official Review · Reviewer_somb · 2024-11-02

**Soundness:** 3
**Presentation:** 3
**Contribution:** 3
**Rating:** 6
**Confidence:** 3

**Summary:**

This paper tries to extend medical LLMs into different languages. The authors constructed a high-quality medical dataset and proposed a novel MoE routing method that employs language-specific experts and cross-lingual routing. The intuition of extending LLMs into different languages is very important, and the dataset is useful in the community.
Several concerns about this research:
1. The paper focused on the medical LLMs, while the evaluation is based on MMLU; although it includes part of the medical-related tasks, the evaluation on this general domain dataset will bring additional noise on evaluating LLMs in medical tasks. More experiments on medical-specific benchmarks should be helpful to demonstrate the contribution of the medical dataset. Otherwise, we can only see the constructed dataset can contribute to the general domain but not specifically to the medical domain.
2. Much relevant research on extending pre-trained models to different languages is overlooked in this paper (related works). For example,  "GreenPLM: Cross-Lingual Transfer of Monolingual Pre-Trained Language Models at Almost No Cost, IJCAI 2023" explores a simple way to extend LLMs to different languages. It would be better to compare the proposed method with other language extension methods.

**Strengths:**

This paper tries to extend medical LLMs into different languages. The authors constructed a high-quality medical dataset and proposed a novel MoE routing method that employs language-specific experts and cross-lingual routing. The intuition of extending LLMs into different languages is very important, and the dataset is useful in the community.

**Weaknesses:**

Several concerns about this research:
1. The paper focused on the medical LLMs, while the evaluation is based on MMLU; although it includes part of the medical-related tasks, the evaluation on this general domain dataset will bring additional noise on evaluating LLMs in medical tasks. More experiments on medical-specific benchmarks should be helpful to demonstrate the contribution of the medical dataset. Otherwise, we can only see the constructed dataset can contribute to the general domain but not specifically to the medical domain.
2. Much relevant research on extending pre-trained models to different languages is overlooked in this paper (related works). For example,  "GreenPLM: Cross-Lingual Transfer of Monolingual Pre-Trained Language Models at Almost No Cost, IJCAI 2023" explores a simple way to extend LLMs to different languages. It would be better to compare the proposed method with other language extension methods.

**Questions:**

Typo: "shwows"

---

> ### Author Response · Authors · 2024-11-25
> **Thanks for your kind reviews**
>
> Thank you for your kind and insightful comments. This paper aims to democratize medical AI for a broader linguistic community. By collecting data, analyzing and leveraging the model's multilingual mechanisms, and integrating linguistic priors, it provides the community with high-quality data, efficient, generalizable, and scalable technical solutions, as well as high-performing models. Below, we will address your questions one by one.
>
> > **Q1: The paper focused on the medical LLMs, while the evaluation is based on MMLU; although it includes part of the medical-related tasks, the evaluation on this general domain dataset will bring additional noise on evaluating LLMs in medical tasks.**
>
> We sincerely apologize for any misunderstanding caused by unclear writing. Regarding the evaluation, we only utilized the **medical-related sections of MMLU** to create evaluation sets for languages without existing medical evaluation datasets. For other languages, we used **peer-reviewed medical evaluation datasets**, which are detailed in **Appendix A.3** and **A.5** and referenced in lines 127-128 and 225-226 of the original paper. In total, we integrated **10** medical evaluation datasets for the corresponding languages and supplemented others with the medical section of MMLU for other languages. The specific datasets are described in detail in **Table 7** of the original paper.
>
>
>
> > **Q2: Much relevant research on extending pre-trained models to different languages is overlooked in this paper (related works).**
>
> GreenPLM is an impressive contribution, sharing the same motivation as our work—to efficiently expand the model's multilingual capabilities. Since many similar outstanding works focus on optimizations based on BERT, they were not included in the related work section. Following your suggestion, we have supplement the related work to provide readers with a broader perspective and highlight the substantial groundwork and innovative ideas within the multilingual community. Please refer to lines 451-458 in the revised version for details.
>
>
>
> **Typo: "shwows"** We have fixed it in the new version. Thank you for carefully reviewing our work.

---

> ### Author Response · Authors · 2024-11-29
> **Follow-Up on Review and Feedback**
>
> Dear Reviewer **somb**,
>
> We hope this message finds you well.
>
> We have carefully addressed all your questions and concerns, including conducting additional experiments as requested, and have provided detailed responses in the rebuttal.
>
> As the rebuttal deadline is approaching, we would deeply appreciate it if you could share your updated thoughts based on the rebuttal and paper revision, or do not hesitate to let us know if you have additional questions, and we will respond promptly.
>
> Thank you again for your thoughtful review and your invaluable contributions to the quality of this paper.
>
> Kind regards,
>
> Paper 10250 Authors

---

> > ### Comment · Reviewer_somb · 2024-11-29
> >
> > I have no further questions

---

> > > ### Author Response · Authors · 2024-11-30
> > >
> > > Thanks for your kind reviews.

---

### Official Review · Reviewer_kMZh · 2024-11-04

**Soundness:** 3
**Presentation:** 2
**Contribution:** 3
**Rating:** 6
**Confidence:** 4

**Summary:**

This paper addresses the challenge of adapting medical large language models (LLMs) to low-resource languages, where data scarcity hinders equitable access to healthcare information.
The authors construct a high-quality medical dataset covering 12 languages and develop a novel approach to model scalability using Mixture of Experts (MoE) modularity and a new routing method.
Their analysis reveals a “Spread Out in the End” information flow mechanism, where cross-lingual information is focused in early model layers but diverges into language-specific pathways in later layers.
This insight leads to the development of the Post-MoE architecture, which applies sparse routing in later layers to enhance model generalization and interoperability.
Furthermore, by introducing "language family" experts, the model efficiently scales to 50 languages without adding extra parameters, making it both scalable and efficient for multilingual medical applications.

**Strengths:**

* This paper tackles the crucial issue of reducing language-related barriers to healthcare access for minor languages through LLMs.

* This paper proposes an intuitive and effective MoE approach for supporting multiple languages within a single MoE model.

* This paper provides an analysis of the inner flow of MoE gating.

* The experimental results are encouraging within the evaluation framework applied in this paper.

**Weaknesses:**

* The core idea of the proposed method does not appear particularly novel, as it resembles a combination of BTX and Sperse Upcycling, as discussed in the related work section.

* Although the results of the proposed method are consistently better than those of the baseline models with similar model sizes for dense models, MoE models have more parameters than the baseline dense models. Therefore, it may be unfair to directly compare the results between the baseline dense model and the MoE model with the same base model size. The authors should also provide results for dense models with model sizes comparable to those of the MoE models obtained by the proposed method.

**Questions:**

* According to Figure 2, Does the base model for the proposed method use a post-normalization Transformer instead of a pre-normalization one?


* This paper claims, "A major obstacle is the scarcity of medical data in many languages, limiting model development for underrepresented populations." According to Table 4, the average accuracies of major and minor languages among 50 languages in GPT-4o are 85.7 and 81.1, respectively. It appears that the performance of minor languages is not significantly worse compared to major languages in recent top models. This result seems to contradict the situation described in the paper. Do the authors provide any other explanation for the motivation behind developing the proposed method?


* The proposed method's best results, namely, an average accuracy of 69.9 for major languages and 58.3 for minor languages, as shown in Table 5, do not seem significantly better than the "dense" baseline, which achieved 69.0 and 55.7, respectively. Since I'm not familiar with the evaluation dataset used in this paper, I wonder if the differences of 0.9 and 0.6 for major and minor languages, respectively, are meaningful enough to consider.

---

> ### Author Response · Authors · 2024-11-25
> **Thanks for your kind reviews (1/N)**
>
> Thank you for your kind and insightful comments. This paper aims to democratize medical AI for a broader linguistic community. By collecting data, analyzing and leveraging the model's multilingual mechanisms, and integrating linguistic priors, it provides the community with high-quality data, efficient, generalizable, and scalable technical solutions, as well as high-performing models. Below, we will address your questions one by one.
>
> > **Q1: The core idea of the proposed method does not appear particularly novel**
>
> From the perspective of the Hybrid Routing method, this approach addresses the issue in the BTX method where the initialization of the MoE model's Attention layer cannot directly align with the Attention layers of expert models trained on specific tasks. Additionally, it provides interpretability that Sparse Upcycling lacks.
>
> More importantly, from the methodological perspective of the paper, we view this approach as a tool for exploring model mechanisms. By employing Post-MoE and introducing language-family experts, we fully leverage the discovered mechanisms to efficiently achieve the original objectives.
>
>
>
> > **Q2: It may be unfair to directly compare the results between the baseline dense model and the MoE model with the same base model size.**
>
> Thank you for your suggestion, which will indeed enhance the robustness of the paper. We have conducted additional experiments based on your recommendations and will present them below.
>
> **Experiment Settings**: To accurately construct a dense model of equivalent size, we replicated the MLP following the approach used in MoE Upcycling and initialized the routing with an average distribution. Unlike MoE, the initialized dense model employs full activation instead of sparse activation.
>
> - **Supplementary Experiments Related to Table 2**:
>
>   The purpose of Table 2 is to demonstrate the generalization advantage of the Hybrid Routing method over other approaches. We have provided **supplementary baseline results** for dense models with parameters matching either the total parameters and the activation parameters of the MoE models. As shown in the table, compared to a Dense model with the same total parameters, **the MoE model exhibits better generalization** on minor languages. Additionally, compared to a Dense model with the same activation parameters, **the MoE model performs better** on both major and minor languages. Please refer to lines 195, 222-224 in the revised version for details.
>
>   | Method/Model                   | Active Param. | Param.    | Avg. Major | Avg. Minor |
>   | ------------------------------ | ------------- | --------- | ---------- | ---------- |
>   | Qwen2-0.5B                     | 0.49B         | 0.49B     | 29.7       | 31.5       |
>   | Dense                          | 0.49B         | 0.49B     | 37.8       | 24.6       |
>   | Dense with Same  Active Param. | **0.81B**     | 0.81B     | 38.4       | 26.2       |
>   | Dense with Same Total Param.   | 3.95B         | **3.95B** | **42.0**   | 30.9       |
>   | MoE with Hybrid-k Routing      | **0.81B**     | **3.95B** | 40.0       | **32.0**   |
>
> - **Supplementary Experiments related to Table 5**:
>
>   The purpose of Table 5 is to evaluate the scalability of the primary method proposed in this paper (PostMoE with language family experts). Due to time constraints, we only supplemented the dense baselines with the same activation parameters. As shown in the Table, the **MoE models outperform the Dense models** with the same number of activation parameters. Please refer to lines 394-410 in the revised version for more details.
>
>   | Method/BaseModel               | Active Param. | Avg. Major | Avg. Minor |
>   | ------------------------------ | ------------- | ---------- | ---------- |
>   | **Qwen2-0.5B**                 | 0.49B         | 29.7       | 31.5       |
>   | Dense                          | 0.49B         | 39.2       | 32.2       |
>   | Dense with Same  Active Param. | 0.52B         | 39.4       | 34.0       |
>   | MoE with Hybrid-k Routing      | 0.52B         | **40.5**   | **34.6**   |
>   | &nbsp;                         |               |            |            |
>   | **Qwen2-1.5B**                 | 1.54B         | 42.9       | 38.4       |
>   | Dense                          | 1.54B         | 52.2       | 43.7       |
>   | Dense with Same  Active Param. | 1.63B         | 52.8       | 44.1       |
>   | MoE with Hybrid-k Routing      | 1.63B         | **54.8**   | **44.9**   |
>   | &nbsp;                         |               |            |            |
>   | **Qwen2-7B**                   | 7.62B         | 55.2       | 49.2       |
>   | Dense                          | 7.62B         | 69.0       | 55.7       |
>   | Dense with Same  Active Param. | 8.02B         | 68.5       | 56.3       |
>   | MoE with Hybrid-k Routing      | 8.02B         | **69.9**   | **58.3**   |

---

> ### Author Response · Authors · 2024-11-25
> **Thanks for your kind reviews (2/N)**
>
> > **Q3: It appears that the performance of minor languages is not significantly worse compared to major languages in recent top models. Do the authors provide any other explanation for the motivation behind developing the proposed method?**
>
> **Contribution to the Open Source Community:** For open-source models, the accuracy gap between major and minor languages exceeds 10% and the ACC is under 50, underscoring a significant disparity in this domain within the open-source community. Our work open-sources all data, code, and models, providing the community with an efficient and practical solution to address this issue.
>
> **Additional Motivations:**
>
> - **Privacy Considerations:** In medical services, a locally deployed medical AI model is essential to eliminate the risk of personal medical data leakage. Current closed-source AI service cannot address this issue.
> - **Community Accessibility:** In many minor language communities, particularly in regions with limited medical resources, Internet access is often challenging. A locally operable medical AI model can mitigate this issue at a very low cost. Current closed-source AI service cannot address this issue.

---

> ### Author Response · Authors · 2024-11-25
> **Thanks for your kind reviews (3/N)**
>
> > **Q4: I wonder if the differences of 0.9 and 0.6 for major and minor languages, respectively, are meaningful enough to consider.**
>
> First, for the 7B model, the performance improved of major and minor languages are 0.9 and **2.6**, respectively, with a competitive increase in the average performance of minor languages.
>
> More importantly, the primary method proposed in this paper (PostMoE with language family experts) offers **the advantages of generalizability**, enabling adding other languages efficiently and effectively. To provide a more intuitive understanding, we selected two additional languages not included in the 50 languages, "Bengali" (Bn) and "Amharic" (Am), to demonstrate the model's efficient generalization capability.
>
> Specifically, we processed 2,000 data samples for "Bengali" (Bn) and "Amharic" (Am) and continued fine-tuning the Dense models and ApolloMoE models. The results of different scales are shown in the tables below.
>
> The experimental results demonstrate the clear advantages of the proposed method in adapting to additional languages. For **performance improvement in newly added languages**, the ApolloMoE model outperforms the Dense model across all scales, with training-related gains also showing advantages in most cases. For **preserving the performance of original languages**, the ApolloMoE model maintains or even improves performance across nearly all scales, whereas the Dense model generally experiences a decline in performance. Please refer to lines 363-365 and  App.E of revised version for more details.
>
> | Method/BaseModel                             | Avg. Major      | Avg. Minor      | Bn                  | Am                  |
> | -------------------------------------------- | --------------- | --------------- | ------------------- | ------------------- |
> | **Qwen2-0.5B Trained on 50 Languages' Data** | 39.2            | 33.2            | 36.4                | 31.5                |
> | Continued Fine-tuning on Bn Data             | 38.2 (-1.0)     | 33.1 (-0.1)     | 37.5 (+0.9)         |                     |
> | Continued Fine-tuning on Am  Data            | 38.1 (-1.1)     | 32.9 (-0.3)     |                     | 33.9 (+2.4)         |
> | **ApolloMoE (From Qwen2-0.5B)**              | 40.5            | 34.6            | **37.8**            | **33.0**            |
> | Continued Fine-tuning on Bn Data             | 40.6 **(+0.1)** | 34.6 **(-0.0)** | **39.7** **(+1.9)** |                     |
> | Continued Fine-tuning on Am  Data            | 40.5 **(-0.0)** | 34.6 **(-0.0)** |                     | **36.0** **(+3.0)** |
>
> &emsp;
>
> | Method/BaseModel                             | Avg. Major      | Avg. Minor      | Bn              | Am                  |
> | -------------------------------------------- | --------------- | --------------- | --------------- | ------------------- |
> | **Qwen2-1.5B Trained on 50 Languages' Data** | 52.2            | 43.7            | 44.0            | 35.0                |
> | Continued Fine-tuning on Bn Data             | 49.1 (-3.1)     | 39.2 (-4.5)     | 50.8 **(+6.8)** |                     |
> | Continued Fine-tuning on Am  Data            | 47.7 (-4.5)     | 39.1 (-4.6)     |                 | 36.4 (+1.4)         |
> | **ApolloMoE (From Qwen2-1.5B)**              | 54.8            | 44.9            | **50.4**        | **38.6**            |
> | Continued Fine-tuning on Bn Data             | 54.0 (**-0.8**) | 44.9 **(-0.0)** | **55.7** (+5.3) |                     |
> | Continued Fine-tuning on Am  Data            | 54.8 (**-0.0**) | 44.9 **(-0.0)** |                 | **41.9** (**+3.3**) |
>
>
> &emsp;
>
>
> | Method/BaseModel                           | Avg. Major      | Avg. Minor      | Bn              | Am              |
> | ------------------------------------------ | --------------- | --------------- | --------------- | --------------- |
> | **Qwen2-7B Trained on 50 Languages' Data** | 69.0            | 56.7            | 66.3            | 35.7            |
> | Continued Fine-tuning on Bn Data           | 68.4 (-1.0)     | 55.7 (-1.0)     | 68.9 **(+2.6)** |                 |
> | Continued Fine-tuning on Am  Data          | 68.3 (-1.1)     | 55.3 (-1.4)     |                 | 38.2 (+2.5)     |
> | **ApolloMoE (From Qwen2-7B)**              | 69.9            | 58.3            | **67.1**        | **40.5**        |
> | Continued Fine-tuning on Bn Data           | 69.6 **(-0.3)** | 58.5 **(+0.2)** | **69.5** (+2.4) |                 |
> | Continued Fine-tuning on Am  Data          | 69.5 **(-0.4)** | 58.3 **(-0.0)** |                 | **42.5 (+2.5)** |
>
> &emsp;
>
> **Typo. Figure2**: We have corrected the image. For further details, please refer to lines 170-179 of the revised version. Thank you for your careful review.

---

> > ### Comment · Reviewer_kMZh · 2024-12-03
> >
> > Yes, 0.6 is a typo and should actually be 2.6. I apologize for the oversight. However, the typo does not change the essence of my question, and I am satisfied with the author's response.
> >
> >
> > **Simple clarification**: The response "Typo. Figure2: We have corrected the image. For further details, please refer to lines 170-179 of the revised version. Thank you for your careful review." is the answer to my question: "According to Figure 2, does the base model for the proposed method use a post-normalization Transformer instead of a pre-normalization one?" since it seems that the current revised version has updated Figure 2 to depict a pre-normalization model. Is my understanding correct?

---

> > > ### Author Response · Authors · 2024-12-03
> > >
> > > Yes. In the current revised version, we have updated Figure 2 to depict a pre-normalization model. Thanks for your careful review.

---

> ### Author Response · Authors · 2024-11-29
> **Follow-Up on Review and Feedback**
>
> Dear Reviewer **kMZh**,
>
> We hope this message finds you well.
>
> We have carefully addressed all your questions and concerns, including conducting additional experiments as requested, and have provided detailed responses in the rebuttal.
>
> As the rebuttal deadline is approaching, we would deeply appreciate it if you could share your updated thoughts based on the rebuttal and paper revision, or do not hesitate to let us know if you have additional questions, and we will respond promptly.
>
> Thank you again for your thoughtful review and your invaluable contributions to the quality of this paper.
>
> Kind regards,
>
> Paper 10250 Authors

---

> ### Author Response · Authors · 2024-12-02
>
> Dear Reviewer kMZh,
>
> We hope this message finds you well.
>
> We have carefully addressed all your questions and concerns, including conducting additional experiments as requested, and have provided detailed responses in the rebuttal.
>
> As the rebuttal deadline is approaching, we would deeply appreciate it if you could share your updated thoughts based on the rebuttal and paper revision, or do not hesitate to let us know if you have additional questions, and we will respond promptly.
>
> Thank you again for your thoughtful review and your invaluable contributions to the quality of this paper.
>
> Kind regards,
>
> Paper 10250 Authors

---

> ### Comment · Reviewer_kMZh · 2024-12-03
>
> (I sincerely apologize for the delay in my response.)
> I would like to thank the authors for their thorough efforts in addressing my questions and concerns, including conducting additional experiments.
> Since most of my concerns have been sufficiently addressed, I have increased my score and voted to accept the submission to the conference.

---

> > ### Author Response · Authors · 2024-12-03
> >
> > Thanks for your kind reviews.

---

### Author Response · Authors · 2024-11-27
**General Response**

Dear Reviewers and ACs,

We sincerely thank all the reviewers and ACs for your diligent efforts and high-quality reviews. **If you have any additional questions or require further clarification, please feel free to let us know. Your insights are highly valued.**

We are delighted to note that reviewers find that:

- Our approach to reducing language-related barriers to healthcare access for minor languages through MoE models is both impactful and effective, with promising results in scalability and performance (Reviewers `kMZh`, `sQUP`, `somb`).
- Our detailed analysis of MoE gating, cross-lingual routing, and the "Spread Out in the End" phenomenon enhances both the interpretability and efficiency of multilingual models (Reviewers `kMZh`, `sQUP`, `CPZy`).
- The high-quality design choices, thorough experiments, and the new model for the medical domain are well-received and contribute significantly to the field (Reviewers `CPZy`, `somb`).

In response to your valuable suggestions, we have conducted additional experiments and made the following modifications in the Rebuttal-PDF for your convenience:

- Table 2: We introduced dense baselines with identical activation parameters and total parameters as the MoE model (as suggested by Reviewers `kMZh` and `sQUP`).

- Figure 2: We corrected the diagram's error concerning the Norm block (suggested by Reviewer `kMZh`).

- Figure 3: The image resolution was enhanced to provide a better viewing experience for readers (suggested by Reviewer `sQUP`).

- Table 5: Dense baselines with matching activation parameters to the MoE models were added (as recommended by Reviewers `kMZh` and `sQUP`).

- Related Work: We incorporated references to studies focused on enhancing multilingual capabilities (suggested by Reviewer `somb`).

- Table 9, App.A.6: MMMLU evaluation data and Manual detection experiments were added to substantiate the validity of the translation data (suggested by Reviewers `sQUP` and `CPZy`).

- Table 10: We included experiments demonstrating the generalization performance of the ApolloMoE model in comparison to dense models (suggested by Reviewer `kMZh`).



Best regards,

The Authors

---

### Meta-Review · Area_Chair_JGwe · 2024-12-20

**Metareview:**

This paper tackles the task of adapting medical LLMs to new languages in order to increase access to these technologies. To do this, the work first introduces a new medical QA dataset for model tuning in 12 languages; this dataset is automatically converted from text to QA format with ChatGPT and then used to finetune a dense LM. They then scale up their setting using a novel MoE architecture (Post-MoE) and routing method, based on findings from their analysis of multilingual information flow within the models. The multilingual medical MoE model is shown to work well in both "major" (I assume this means high-resource?) and "minor", or low-resource, languages, suggesting that this is a promising direction for multilingual adaptation of LLMs.

Strengths:
- The paper addresses an important issue (kMZ, somb) and provides a well-designed approach for tackling it with MoE models (kMZ, CPZy), as well as a detailed analysis of the method to better understand how it works (kMZh, CPZy).
- The authors construct a new medical dataset for model tuning, which fills a currently open niche in terms of instruction tuning datasets (somb).
- The paper is well-written and easy to follow (CPZy).
- Comprehensive experiments (sQUP), particularly after adding many experiments to address reviewers' concerns, including more evaluations using a new dataset, MMLU, and additional baselines with comparable parameter counts to the MoE models.

Weaknesses:
 - The experimental results show relatively small gains in performance, even for minor/low-resource languages (kMZh). The gains across languages are uneven (some languages benefit much more than others) without an analysis of why this occurs (sQUP).
 - Similarly, the grouping of languages by "language family" is quite coarse-grained and doesn't take into account the many factors (script, grammatical similarity, vocabulary overlap, etc.) that are known to affect cross-lingual learning in LMs (e.g., [1]). The scaled model would likely work much better, particularly for the low-resource languages, if languages were more carefully selected and grouped into experts.
 - A minor note, but using "major" and "minor" as language descriptors is uncommon in multilingual research. I recommend changing this to be "high-" and "low-resource" languages, or adding an explanation for why this terminology is used instead.

[1] Chang et al., When Is Multilinguality a Curse? Language Modeling for 250 High- and Low-Resource Languages. https://arxiv.org/abs/2311.09205

**Additional Comments On Reviewer Discussion:**

The authors provided a detailed response that addressed most concerns and added appropriate new experiments to the paper. In response, one reviewer raised their score to recommend acceptance.

---

### Decision · Program_Chairs · 2025-01-22

Accept (Poster)